

# Destruction of localization by thermal inclusions: Anomalous transport and Griffiths effects in the Anderson and André-Aubry-Harper models

Xhek Turkeshi[1,†], Damien Barbier[2,3,†],
Leticia F. Cugliandolo[2,5], Marco Schirò[1] and Marco Tarzia[4,5,⋆]

**1** JEIP, USR 3573 CNRS, Collège de France, PSL Research University,
11 Place Marcelin Berthelot, 75321 Paris Cedex 05, France
**2** Sorbonne Université, Laboratoire de Physique Théorique et Hautes Energies, CNRS UMR
7589, 4 Place Jussieu, Tour 13, 5ème étage, 75252 Paris 05 France
**3** Ecole Polytechnique Fédérale de Lausanne, Information, Learning and Physics lab and
Statistical Physics of Computation lab, CH-1015 Lausanne, Switzerland
**4** Sorbonne Université, Laboratoire de Physique Théorique de la Matière Condensée, CNRS
UMR 7600, 4 Place Jussieu, Tour 13, 5ème étage, 75252 Paris Cedex 05, France
**5** Institut Universitaire de France, 1 rue Descartes, 75231 Paris Cedex 05 France

⋆ marco.tarzia@gmail.com
† Equal contribution

## Abstract

We discuss and compare two recently proposed toy models for anomalous transport and Griffiths effects in random systems near the Many-Body Localization transitions: the random dephasing model, which adds thermal inclusions in an Anderson Insulator as local Markovian dephasing channels that heat up the system, and the random Gaussian Orthogonal Ensemble (GOE) approach which models them in terms of ensembles of random regular graphs. For these two settings we discuss and compare transport and dissipative properties and their statistics. We show that both types of dissipation lead to similar Griffiths-like phenomenology, with the GOE bath being less effective in thermalising the system due to its finite bandwidth. We then extend these models to the case of a quasi-periodic potential as described by the André-Aubry-Harper model coupled to random thermal inclusions, that we show to display, for large strength of the quasiperiodic potential, a similar phenomenology to the one of the purely random case. In particular, we show the emergence of subdiffusive transport and broad statistics of the local density of states, suggestive of Griffiths like effects arising from the interplay between quasiperiodic localization and random coupling to the baths.

# 1 Introduction

Transport properties of quantum many body systems are known to be extremely sensitive to inhomogeneities and spatial disorder. A well understood example is provided by the phenomenon of Anderson localization [1], where diffusion is suppressed by quantum interference above a critical disorder strength, which turns out to be vanishingly small in low dimensions [2]. Another notable example is provided by the case of a quasi-periodic potential incommensurate with the lattice, as in the model of André-Aubry-Harper (AAH) [3,4] where in one dimension a transition from delocalized ballistic transport to localized behavior arises as a function of the potential strength.

The fate of quantum localization in presence of many-body interactions, and in regimes far from low temperature equilibrium, has been at the center of theoretical and experimental interest, both for its fundamental interest for our basic understanding of quantum statistical

mechanics and for its practical implications in the search for mechanisms to protect quantum information [5–13]. The existence of a Many-Body Localized (MBL) phase has been discussed and established, both for quenched randomness and quasi-periodic disorder [14–20], yet many questions remain open. In particular the transport and thermalization properties on both sides of the MBL transition have attracted particular interest, triggered by the evidence for anomalous (sub) diffusion [21–30], a remarkably robust phenomenon which has been since then also observed experimentally with cold atoms, ultracold ions, and superconducting circuits [31–35].

An appealing phenomenological interpretation of these anomalies has been proposed, for disordered models, in terms of the existence around the MBL transition of a quantum Griffiths phase [36]. This is characterized by rare inclusions of the insulating (conducting) phase with an anomalously small (large) localization length. Several phenomenological proposals have been put forward to describe this physics, from purely classical resistor-capacitor models with power-law distributed resistances [23, 37] to stochastic models for merging thermal and insulating regions which are motivated by strong-disorder renormalization group arguments [38–41]. Crucially, this Griffiths physics is at the core of our understanding of both the critical properties of the MBL transition and the mechanism for its destabilization through quantum avalanches [42–48].

In the quasi-periodic case the understanding of transport and thermalization across the MBL transition is even more preliminary. Anomalous transport has been reported in the past years for the interacting AAH model, including claims of subdiffusion and superdiffusion [49–51], which have been, however, recently questioned [52]. The existence of a subdiffusive region, prior to the fully localized phase, is an intriguing possibility and its relation with a possible Griffiths phase is a priori not obvious. In fact, the quasi-periodicity is a structured pattern, whereas disordered potentials are random and uncorrelated processes. A better understanding of the robustness of quasi-periodic localized phases against thermal inclusions and their transport properties is therefore urgent.

Unfortunately, the study of large disordered or quasi-periodic interacting many body systems with numerically exact methods remains extremely challenging. It is therefore desirable to find models which capture some of the key physical facets of many-body localizing systems, while being computationally tractable even for large system sizes.

Recently, two toy models have been proposed to describe, at a microscopic level, the interplay between localization and the so-called thermal inclusions, which in truly many-body systems are produced by internal interactions, in the context of MBL with random disorder. The first one, called random dephasing model [53], describes the environment in terms of Markovian dephasing processes able to heat up the system and which are randomly coupled to each lattice site with a given probability. The second one describes thermal inclusions as the coupling to a GOE random-matrix bath implemented through an ensemble of Random-Regular-Graphs (RRG) [54]. In addition to their role for the understanding of MBL physics these models also present an intrinsic interest, as simple settings where questions related to stability of localization with respect to dissipation can be answered in some detail.

It is important to mention here that some recent works have even questioned the existence of a truly MBL phase in the thermodynamic limit [55–59]. Furthermore, several recent numerical results indicate that, even if a genuine MBL transition exists, it is affected by strong finite-size effects, with the transition point moving to larger and larger values of the disorder as larger system sizes are considered [60, 61], and eventually following far outside the numerically-accessible crossover between the finite-size MBL regime and thermalization. Yet, the subdiffusive crossover region preceding the transition seem to be very robust and much less affected by finite size effects. In this respect Refs. [62–65] put forward the idea that that the MBL transition and the anomalously slow sub-diffusive crossover phase preceding it may

be driven by distinctly different physical mechanisms. In this paper we focus on this latter regime. In this regard, the advantage of working with simplified toy models is twofold. On the one hand, they allow one to inspect the physics on much larger scales (on which Griffiths effects are believed to be important) compared to the interacting models. On the other hand, they allow one to study the sub-diffusive regime disentangling it from the MBL transition.

In the first part of this work we revisit and compare in detail these two models of thermal inclusions in the context of a simple Anderson insulator. In particular, we compute transport and spectral properties, going beyond the results presented in [53,54]. We then extend these toy models to the quasi-periodic case, with the goal of shedding light on the mechanism for the destruction of their localized phases by thermal inclusions and assessing its degree of universality.

The paper is structured as follows. In Sec. 2 we give an overview of the main results of our paper. In Sec. 3 we define the Anderson and AAH models, the dissipative settings we consider, the main quantities of interest and we also briefly mention how to compute them. In Sec. 4 we present our results for the random disordered case, while in Sec. 5 we discuss the quasi-periodic one. In Sec. 6 we discuss and compare the behavior of the two models and we discuss them in the light of possible connections with the MBL problem. Finally, in Sec. 7 we present our conclusions. We detail in the Appendix the derivations and computations of our analytic results.

## 2   Summary of Main Results

We start with an overview of the main results obtained in this work, which will be discussed in detail in Sections 4 and 5.

We consider two models for single particle localization, namely one dimensional quantum particles in presence of (uncorrelated) random or quasiperiodic disorder leading to the Anderson and AAH models defined in Sec. 3.1, and we study the effect of thermal inclusions on their transport and spectral properties. As we discuss in Sec. 3.2 for each of these two models we consider and compare two types dissipative environments: (i) a Markovian Dephasing Bath (MDB) described by the Lindblad master equation with randomly distributed dephasing jump operators, similar to what was done in Ref. [53], and (ii) a GOE bath described by a local coupling to independent Random-Regular-Graphs, as done in Ref. [54]. The advantage of these two settings is that in both cases the effect of the coupling to the bath can be treated exactly using Lindblad equations of motions or the cavity method, respectively, and these lead to numerically exact results for single-particle observables such as the particle current or the local density of states, as discussed in Sec. 3.3.

We first discuss in Sec. 4 the case of the Anderson model coupled to MDB and GOE baths, for which we revisit and complement the results of Refs. [53,54]. In particular, we discuss transport properties such as the scaling of the typical resistivity with system size, a study not performed in Ref. [54], and the statistics of the Local Density of States (LDoS), missing in Ref. [53], thus obtaining a complete picture of the effect of thermal inclusions on the Anderson localized phase leading to a robust sub-diffusive phase. Our results are summarized in the dynamical phase diagram shown in Fig. 5. This study lets us demonstrate that the two ways of introducing dissipation, MDB and GOE, lead to the same physical behavior. Already from this analysis we can conclude that the GOE bath is less effective in thermalizing the system and inducing diffusion.

In Sec. 5 we extend our analysis to the quasiperiodic case, i.e. the AAH model. We discuss how dissipation in the two settings affects its transport and spectral properties. Our results show that in both cases the localized phase of the AAH model turns into a sub-diffusive regime

which appears to be much broader for the GOE dissipation than for the MDB one, as shown in the dynamical phase diagram of Fig. 9. This subdiffusive scaling appears also in the statistics of the LDoS.

A comment is in order here. For a truly many-body model in a quasiperiodic potential the spatial structure of the thermal inclusions is not random and is arguably correlated with the potential itself. However, predicting the position of these thermal spots is an extremely hard task, which is essentially equivalent to solving the many-body problem. For this reason in the toy models we work with the dephasing dynamics is assumed to apply on a randomly distributed set of sites, independently of the local potential, both in the Anderson and in the AAH case. One could then argue that, since in our simplified setting the insulating regions and the thermal inclusions are essentially put by hand at random, one cannot draw any conclusion on the universality of the Griffiths picture. Yet, on the one hand one could speculate that many-body interactions, together with the randomness of the initial configuration, give rise to some sort of configurational disorder, as one could see for example by treating those interactions at the Hartree-Fock (HF) level [66, 67]. Besides, we would like to point out that the fact that the Anderson model with uncorrelated disorder and the AAH model in the localized regime respond in a very similar way when coupled to a thermalizing system is still interesting and informative. In fact, as explained below, in the Anderson model the formation of rare resonances is a quite "simple" local process: resonances are formed on the sites on which the local disorder is small and the dephasing dynamics is active. Instead the formation of resonances in the AAH case cannot be predicted easily and is likely to involve more complex and non-local processes. Still, the phenomenology of the two models when coupled to the thermal inclusions is essentially the same, for the two kinds of dissipative enviroments that we consider. We believe that this gives some strong indication on the fact that the subdiffusive regime is a very robust feature which appears whenever Anderson localized edgestates are perturbed by thermal inclusions, independently of the details of the microscopic modelization of the bath.

## 3 Models and Methods

In this Section we define the models we work with. We first introduce the two non-interacting one-dimensional chains, namely the Anderson tight-binding model with quenched randomness and the André-Aubry-Harper model with a quasi-periodic potential. Next, we explain the two ways in which we introduce the thermal inclusions due to the coupling to an effective environment. Finally, we identify the observables we employ and we briefly describe how we implement them (a thorough analysis is detailed in the Appendices).

### 3.1 Anderson and André-Aubry-Harper Models

Let us consider a one-dimensional tight-binding model of non-interacting spin-less fermions moving on a chain of $L$ sites. The system's Hamiltonian is

$$\hat{H} = \sum_{i=1}^{L-1} \left[ \mathrm{t}\left(\hat{d}_i^\dagger \hat{d}_{i+1} + \mathrm{h.c.}\right) + h_i \hat{d}_i^\dagger \hat{d}_i \right] = \sum_{i,j=1}^{L} \hat{d}_i^\dagger \mathbb{H}_{i,j} \hat{d}_j \,,$$
$$\mathbb{H}_{i,j} = \mathrm{t}\delta_{i,j+1} + \mathrm{t}\delta_{i+1,j} + h_i \delta_{i,j} \,. \tag{1}$$

In Eq. (1) $\hat{d}_i$ ($\hat{d}_i^\dagger$) are the annihilation (creation) operators acting on the $i$-th site, the one-particle Hamiltonian $\mathbb{H}$ is an $L \times L$ Hermitian and tridiagonal matrix, and open boundary condition are applied. The constant hopping t sets the unit of energy, and the inhomogeneous

potential $h_i$ is taken to be either random and uniformly distributed

$$h_i \in [-W; W],\qquad(2)$$

or quasi-periodic (i.e. incommensurate with the lattice)

$$h_i = \lambda \cos(2\pi \varkappa i),\qquad(3)$$

with $\varkappa = (\sqrt{5}-1)/2$ (known as the golden ratio)[1] and $\lambda \geq 0$. These are the Anderson [1] and the André-Aubry-Harper quasi-periodic [4] models, respectively.

## 3.2  Random Coupling to Baths

With the aim of mimicking the effect of thermal inclusions, we consider a random coupling to baths, modeled in two ways which we specify in the following two subsections. The couplings allow for energy relaxation, with a rate $\gamma_i$ that is a random variable distributed according to

$$P(\gamma_i; p, \gamma) = p\delta(\gamma_i) + (1-p)\delta(\gamma_i - \gamma),\qquad(4)$$

where $p \in [0,1]$ is a control parameter, and $\gamma$ sets the dephasing strength (see the cartoons in Fig. 1 and Fig. 2). In particular, the limits

— $p \to 1$ corresponds to no dephasing at all,

— $p \to 0$ is a constant and uniform dephasing acting on every site.

Given a spin chain of length $L$, $(1-p)L$ is the average number of sites under dephasing. Finally, a reason for setting these coupling terms randomly across the set-up is to mimic the spatial distribution of thermal inclusions in many-body localized systems (at least when considering models with a random potential). For deterministic models the position of a thermal inclusion is not random, although it remains unpredictable. For this reason we keep the random coupling distribution (4) even when focusing on André-Aubry-Harper chains.

### 3.2.1  Markovian Dephasing Bath

In order to include the effect of the environment while keeping the system tractable we work in the framework of open Markovian quantum systems described by the Lindblad equation for the system density matrix [68], i.e.

$$\partial_t \hat{\rho}_t = \mathcal{L}\hat{\rho}_t = -i[\hat{H}, \hat{\rho}_t] + \mathcal{D}[\hat{\rho}_t],\qquad(5)$$

where the first term represents the coherent evolution of the density matrix due to the system's Hamiltonian $\hat{H}$, while the second term accounts for incoherent processes due to the coupling to the environment and it is described by a dissipator $\mathcal{D}[\hat{\rho}_t]$ acting on the density matrix. In this work we consider three types of dissipative processes, which give rise to a dissipator of the form

$$\mathcal{D}[\circ] = \mathcal{D}_{\mathrm{d}}[\circ] + \mathcal{D}_{\mathrm{bnd,l}}[\circ] + \mathcal{D}_{\mathrm{bnd,r}}[\circ],\qquad(6a)$$

$$\mathcal{D}_{\mathrm{d}}[\circ] = \sum_{i=1}^{L} \gamma_i \left(2\hat{n}_i \circ \hat{n}_i - \{\hat{n}_i, \circ\}\right),\qquad(6b)$$

$$\mathcal{D}_{\mathrm{bnd,l}}[\circ] = \Gamma\left(2\hat{d}_1^\dagger \circ \hat{d}_1 - \{\hat{d}_1 \hat{d}_1^\dagger, \circ\}\right),\qquad(6c)$$

$$\mathcal{D}_{\mathrm{bnd,r}}[\circ] = \Gamma\left(2\hat{d}_L \circ \hat{d}_L^\dagger - \{\hat{d}_L^\dagger \hat{d}_L, \circ\}\right).\qquad(6d)$$

---

[1]Any other value which is incommensurate with the lattice spacing, here assumed to be unity, gives equivalent physics. The specific choice of the golden ratio is due to historical reasons.

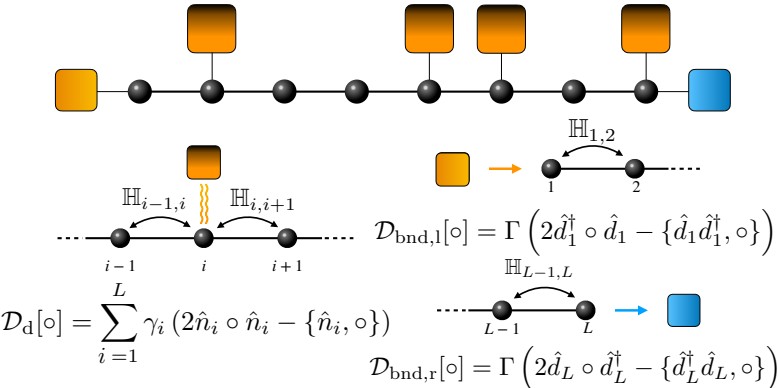

Figure 1: (Top) Cartoon of the Anderson or AAH models randomly coupled to Markovian dephasing baths (MDB). Each black bullet represents a site on the lattice and the red squares above the chain depict the local baths. The absence of some squares illustrates the fact that the coupling to the baths is controlled by the random variable $\gamma_i$ which vanishes with probability $p$. The left and right boundary terms inject/absorb particles from the system. (Bottom) The details of the local couplings on site $i$. $\mathcal{D}_\mathrm{d}$, $\mathcal{D}_\mathrm{bnd,l}$ and $\mathcal{D}_\mathrm{bnd,r}$ represent Lindblad operators in the bulk and at the boundaries.

The first term, given by Eq. (6b), describes so-called dephasing processes (energy exchanges with the bath that do not change particle number) and involve the particle density on each site $i$, $\hat{n}_i \equiv \hat{d}_i^\dagger \hat{d}_i$. As discussed at the beginning of this section, we take the coupling to the bath $\gamma_i$ to be random and distributed according to Eq. (4).

The last two terms, given in Eq. (6c)-(6d), describe particle injection/ejection contributions acting on the first/last site of the chain, respectively. These processes are necessary to induce a direct current (DC) at late times and therefore to describe transport. We take the coupling to these two baths equal and given by $\Gamma$ (see Fig. 1).

For brevity, the above characterization is collectively referred to as Markovian Dephasing Bath (MDB). Within this setup, despite the four body interaction due to the dephasing contribution, we can solve the equations of motion for the quantities of interest exactly [53,69] (see Sec. 3.3). We outline the necessary details in App. A.

### 3.2.2 GOE Bath from Random Regular Graphs

Here we introduce our second setup, which is inspired by the recent proposal of a toy model for the Griffiths phase of the disordered MBL problem [54]. The model is made of $M$ identical copies of Anderson/AAH chains of length $L$, labeled by the index $n = 1, \ldots, M$. To mimic the effect of thermal inclusions, at each horizontal position $i$ the $M$ sites belonging to different chains are coupled by random hopping terms extracted from a sparse random matrix ensemble, i.e., the ensemble of Random Regular Graphs (RRG) of fixed total connectivity $c = 3$ [70]. The RRGs are random lattices with a local tree-like structure, loops with typical length of order $\log M$, and no boundary. It is well known that in the absence of disorder the connectivity matrix of a RRG belongs to the Gaussian Orthogonal Ensemble (GOE) [71,72]. Therefore, in the following we refer to this dissipative setting as GOE bath. We present a sketch of this setup in Fig. 2.

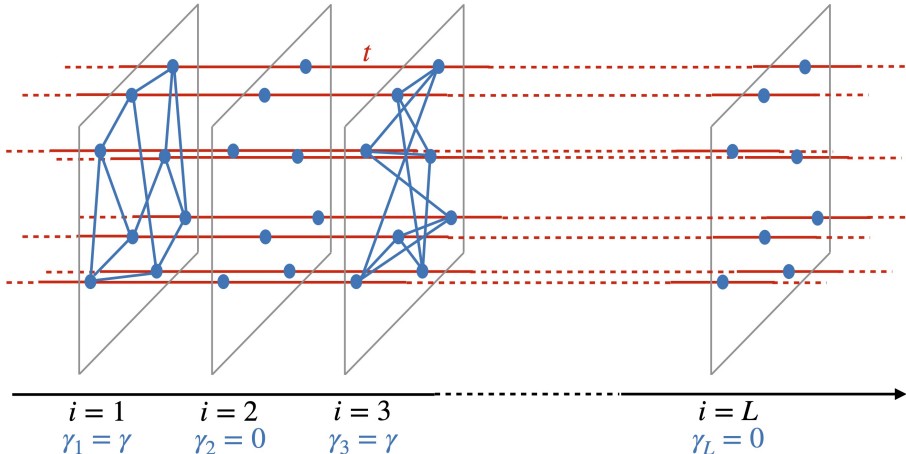

Figure 2: Cartoon of the Anderson or AAH model coupled to random GOE baths with probability $1 - p$. The full model is made of $M$ identical copies of a $1d$ chain coupled along the $x$-direction via the inter-layer hopping rates t (red edges), with potential $h_i$ (equal on all sites of the transverse planes). The matrix elements between two sites at a given position $i$ along the $x$ direction correspond to the adjacency matrix of a random realization of a RRG (different on each layer) times the intra-layer hopping rate $\gamma_i$ (blue edges). The absence of some RRGs illustrates the fact that $\gamma_i = 0$ with probability $p$. In the figure the connectivity of the RRGs is $c = 3$.

The Hamiltonian of the model is

$$
\begin{aligned}
\hat{H}_{\text{GOE}} = &-\sum_{i=1}^{L}\sum_{n=1}^{M}\Big[\mathrm{t}\Big(\hat{d}_{i,n}^{\dagger}\hat{d}_{i+1,n} + \text{h.c.}\Big) + h_i\,\hat{d}_{i,n}^{\dagger}\hat{d}_{i,n}\Big] \\
&-\sum_{i=1}^{L}\sum_{\langle n,m\rangle_i}\gamma_i\Big(\hat{d}_{i,n}^{\dagger}\hat{d}_{i,m} + \text{h.c.}\Big),
\end{aligned}
\tag{7}
$$

where $\hat{d}_{i,n}$ and $\hat{d}_{i,n}^{\dagger}$ are creation and annihilation operators on the site at position $i$ along the $n$-th chain, and $h_i$ is the inhomogeneous potential (which is identical on all $M$ sites sitting at the same position $i$). The latter, in accordance with Sec. 3.2.1, we take to be either uniformly distributed in $h_i \in [-W; W]$ or given by a quasi-periodic incommensurate potential, see Eqs. (2) and (3). The parameter t is the hopping rate in the horizontal direction between sites occupying subsequent positions along the chains. $\gamma_i$ is the hopping rate in the transverse direction and it is equal to $\gamma$ with probability $1 - p$ and zero with probability $p$, as in Eq. (4). The notation $\langle n, m\rangle_i$ indicates pairs of sites $n$ and $m$ with the same horizontal coordinate $i$ connected by an edge of the RRG within the $i$-th plane.

Note that on each vertical plane a different random realization of the RRG is chosen, in such a way that two sites that are connected by $\gamma$ within a given layer are (with high probability in the $M \to \infty$ limit) not connected on another layer. This is important as it ensures that the whole lattice can be thought of as an *anisotropic* random graph, which is locally a tree but has loops whose typical size diverges with the system size. The sites have either connectivity $c + 2$ if they belong to a layer with $\gamma_i \neq 0$ or just 2 in the cases with $\gamma_i = 0$.

The Anderson insulator with $p = 0$, corresponding to a uniform dissipative coupling along the chain, has already been studied in Ref. [54]. In this case one finds a localization/delocalization transition when $\gamma$ becomes of order $1/L$. Unlike the MDB set-up described in Sec. 3.2.1 in which each connection to the dephasing bath produces dissipation, in the GOE set-up the RRG couplings are not always effective to ensure dissipation. This implies that while the average number of non-zero $\gamma_i$ couplings along the chain is $(1 - p)L$, the number of

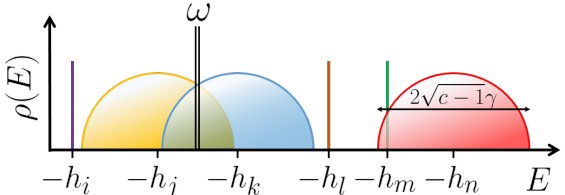

Figure 3: Scheme representing the resonance condition for different layers (labelled $i$, $j$, $k$, $l$, $m$, $n$) with RRG interactions. The density of states in layers $i$, $l$ and $m$ corresponds to a delta peak, which is characterized by an absence of RRG interactions ($\gamma_i = \gamma_l = \gamma_m = 0$). On the contrary for layers $j$, $k$ and $n$ the density of states has a width of $2\sqrt{c-1}\gamma$ as we consider $\gamma_j = \gamma_k = \gamma_n \neq 0$. Layers $j$ and $k$ correspond to the scenario in which a resonance is formed, their density of states is non-zero for $E = \omega$ and consequently it generates a peak in the LDoS around layers $j$ and $k$. For layers $i$, $l$, $m$ and $n$ however we have no overlap between the probing frequency $\omega$ and $\rho(E)$, thus there is no resonance in this case. In general, this means that only the RRG interactions corresponding to the scenario of layers $j$ and $k$ form a resonance in the LDoS.

positions in which the GOE-like couplings produces a Local Density of States (LDoS) of order one only scales as $\sim (1 - p_{\text{eff}})L$, with $p_{\text{eff}} \geq p$. This can be understood with simple arguments in the strong disorder limit ($W \gg$ t or $\lambda \gg$ t depending on the type of on-site potential). When $\gamma = 0$ the system corresponds to $M$ identical Anderson insulators with wave-functions strongly localized around the sites $i$ and corresponding eigenvalues close to the local potential $-h_i$. When the RRG couplings are turned on the hopping in the transverse direction lifts up the degeneracy and spreads these $M$ eigenenergies on a semi-circle-like distribution $\rho(E)$ centered around $-h_i$ and of width $2\sqrt{c-1}\gamma$, see App. B. When the probing energy $\omega$ falls within this band, a resonance in the LDoS is formed on site $i$. On the contrary, if $\omega$ falls outside this band the RRG couplings do not yield a significant contribution to the LDoS on site $i$ (see Fig. 3 for a sketch). Consequently, the fraction of dephasing planes $(1 - p_{\text{eff}})$ is roughly given by the probability to have $\gamma_i \neq 0$ times the probability that the on-site potential $-h_i$ is close enough to the probing energy $\omega$ to form a resonance in the LDoS. This means

$$(1 - p_{\text{eff}}) = (1 - p) \int_{-\infty}^{+\infty} dh_i\, P(h_i)\Theta\left(2\sqrt{c-1}\gamma - |h_i - \omega|\right) < (1 - p)\,, \tag{8}$$

where $P(h_i)$ is the probability distribution of $h_i$ and $\Theta$ the Heaviside function. More generally, this limiting case ($W \gg$ t or $\lambda \gg$ t) is a good example to understand why the RRG interactions act like a bath in this set-up. Indeed, as shown in App. B, the presence of these interactions generates a non-vanishing imaginary part in the Green function $\mathcal{G}(z) = (\hat{H}_{\text{GOE}} - z\hat{\mathbb{1}})^{-1}$, with $z = \omega + i0^+$, provided that there is a resonance. This implies that particles on a given site have a finite life-time as they scatter in the system [73], which in other words means that localization is destroyed. With the RRG interactions we thus retrieve a delocalization phenomenon that is usually induced by a bath acting on a system with localized eigenstates.

## 3.3 Observables

In the following we characterize transport and spectral properties of the systems, which are encoded in the frequency-resolved single particle retarded Green's functions whose imaginary part gives the Local Density of States (LDoS). For transport, a natural observable to distinguish between localized, diffusive or intermediate regimes is the resistance of a finite-sized sample, defined as the inverse stationary current flowing through the system,

$$R \equiv 1/j_\infty\,, \tag{9}$$

which in the stationary ($t \to \infty$) regime is constant throughout the chain for both protocols (see the App. A).

In the MDB case the average current is defined as

$$j_\infty^{\text{MDB}} = i\text{t}\,\langle \hat{d}_{i-1}^\dagger \hat{d}_i - \hat{d}_i^\dagger \hat{d}_{i-1}\rangle_\infty\,. \tag{10}$$

The asymptotic current is expected to be independent of $i$. In the equation above we have introduced the average over the stationary state (infinite time solution of Eq. (5)) $\langle \circ \rangle_\infty \equiv \text{tr}(\circ \rho_\infty)$.

In the GOE setup instead we directly measure the (zero temperature) dimensionless conductance $1/R^{\text{GOE}}$ at a given energy energy $\omega$ through the Fisher-Lee formula [74],

$$1/R^{\text{GOE}}(\omega) = \text{Tr}\left[\Gamma_L G^{\text{GOE}} \Gamma_R \left(G^{\text{GOE}}\right)^\star\right]\,, \tag{11}$$

which uses the retarded (resp. advanced) Green's function $G^{\text{GOE}}(\omega)$ (resp. $(G^{\text{GOE}}(\omega))^\star$) of the system dressed by the leads, and $\Gamma_{L,R} = -2\text{Im}\Sigma_{L,R}$ as the imaginary part of the self-energies associated with the semi-infinite left and right leads (see App. B for details). Because the GOE case involves the probing energy $\omega$, it is important to note that $R^{\text{MDB}}$ and $R^{\text{GOE}}(\omega)$ are not equivalent: $R^{\text{MDB}}$ probes the whole spectrum of the system while $R^{\text{GOE}}(\omega)$ only characterizes the setup at a given energy. Throughout the rest of the paper we will measure the conductance at $\omega = 0$, around the middle of the band of single-particle eigenstates.

Another relevant quantity for our analysis is the single particle retarded Green's function, whose imaginary diagonal part is proportional to the LDoS. Such quantity contains information on the degree of dissipation throughout the system and plays an important role in the theory of localization. We can define the retarded Green's function for the two settings as

$$\text{MDB}: \qquad G_{i,j}^{\text{MDB}}(t) = -i\theta(t)\langle\{\hat{d}_i(t), \hat{d}_j^\dagger\}\rangle\,, \tag{12}$$

$$\text{GOE}: \qquad G_{(i,n),(j,m)}^{\text{GOE}}(z) = \langle i,n|\frac{1}{\hat{H}_{\text{GOE}} - z\hat{\mathbb{I}}}|j,m\rangle\,, \tag{13}$$

respectively. In the MDB setting we have used the time-domain definition of the retarded Green's function, written in terms of the anticommutator of time-evolved fermionic operators and with the theta function ensuring causality. We note that in the context of fermionic open quantum systems the time-evolution has to be performed with the adjoint Lindblad operator as we discuss in App. A.2. In the GOE case it is more convenient to work directly in the frequency representation, $z = \omega + i0^+$, and define the retarded Green's function as the resolvent of the Hamiltonian. As mentioned above, the imaginary part of the retarded Green's function has a direct physical interpretation as LDoS at energy $\omega$, and we study it below. In more explicit form it reads

$$A_{i,n}^{\text{GOE}}(\omega) = \frac{1}{\pi}\text{Im}G_{(i,n),(i,n)}^{\text{GOE}} = \sum_\alpha |\psi_\alpha(i,n)|^2\,\delta(E_\alpha - \omega)\,. \tag{14}$$

Thanks to the translational invariance within the transverse planes in the GOE setting, the dependence of the Green's functions on the in-layer $n$ indices disappears in the $M \to \infty$ limit. In fact the typical length of the loops in the transverse planes diverges as $\log M$. Therefore, since the local potential is the same on each node of the transverse layer, they all become statistically equivalent and the translational invariance in the transverse direction is recovered. An equivalent expression to the one in Eq. (14) can be obtained for the MDB setting (see *e.g.* [75]).

We note that both the resistance $R$ and the retarded Green's functions $G$ defined above are stochastic variables due to the random coupling to the bath for $0 < p < 1$, and have an additional source of randomness for the Anderson model, *i.e.* the quenched disorder potential $h_i$.

We thus study the full probability distribution of these observables. Since the resistance develops a broad distribution with fat tails, $P(R) \sim R^{-\mu}$, approaching the subdiffusive regime [53], we use its typical value as a proxy characterizing the transport properties. Denoting the average over all sources of disorder as $\overline{\circ}$, we specifically define

$$R_{\text{typ}} = \exp \overline{\log R}, \tag{15}$$

where we occasionally append the superscript MDB/GOE specifying the protocol under consideration.[2] We also study the exponent $\mu$ with which $P(R)$ decays at large $R$.

We conclude this section with some technical insights on how the calculation of the relevant observables defined above is performed within the two dissipative settings, leaving further details to the Appendices.

In the MDB setting we can compute both the stationary current as well as the retarded Green's function exactly using the equation of motion techniques, as discussed in App. A. This is possible despite the fact that the dephasing jump operators are proportional to the particle density, thus leading to a Lindbladian which is not quadratic in the fermionic operators. The structure of the dissipator however allows for further simplifications: one can indeed show that the equations of motion for single particle correlations, usually coupled to higher order correlators through a full hierarchy, decouple for the dephasing model [53,69,76]. As a result one can obtain closed equations of motion for both single-time and two times single particle correlations, which can be solved in real-space for generic inhomogeneous couplings. Using this approach we can therefore compute both the steady-state current and the LDoS for the MDB, as we review in App. A.

In the RRG/GOE context, we used two approaches. The first one is based on the Cavity Method and has already been explained in Ref. [54], see also App. B. It consists in taking the limit $M \to \infty$ from the start and assuming that loops are so long that they can be neglected. This method is appropriate to compute local observables such as the probability distribution of the LDoS, and leads to recursion relations for the diagonal elements of the Green's functions (given in App. B) which can be easily solved for very long chains.

A second complementary strategy, also explained in App. B, consists in studying the model at finite $M$ coupled to biased reservoirs at its edges and using a relation between the conductivity and the transmission matrix proposed in [74,77]. The coupling to the semi-infinite left and right leads effectively yields a quasi-1$d$ model, which can be solved exactly with the Transfer Matrix method by inverting the full Green's function within each plane by lower-upper (LU) decomposition [78]. This method is appropriate to compute transport properties and global observables such as the conductivity, as it allows one to overcome the drawback of the first approach, which does not account in an exact fashion for all possible paths joining two sites of the first and last external planes which are attached to the leads. However, the exact recursion equations at finite $M$ are much more costly to solve numerically (the time scales as $LM^3$) and we are thus limited to much smaller sizes, $L \lesssim 256$, compared to the first strategy. Moreover, keeping $M$ finite leads to extra finite-size effects in the transverse direction.

# 4 The Anderson Chain

We start the discussion of our results focusing on the Anderson model coupled to local thermalizing environments, in the spirit of Refs. [53,54]. We first briefly revisit its transport properties and we display and discuss the numerical evidence for sub-diffusion. We then present results

---

[2]Similar results to the ones presented in this paper are obtained when substituting the median med($R$) to the typical value Eq. (15). We do not show them here for presentation convenience.

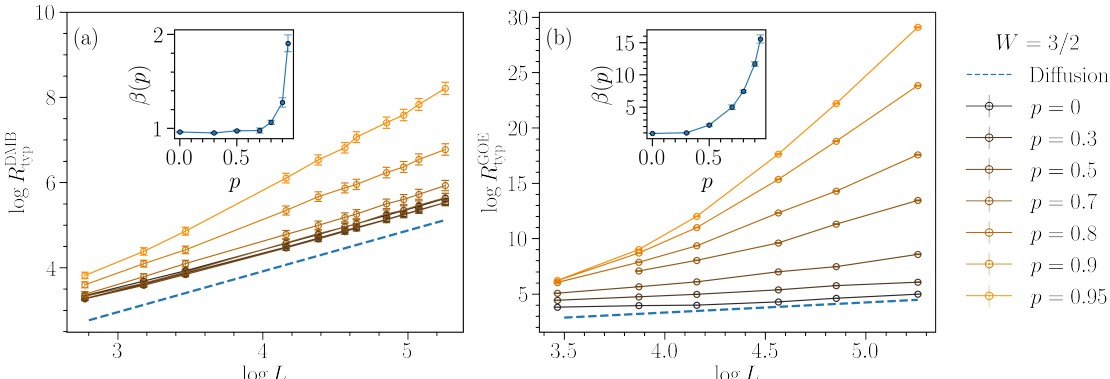

Figure 4: Typical resistance $R_{\text{typ}}$ (cfr. Eq. (15)) at $W = 3/2$ as a function of system size for the MDB (a) and GOE (b) Anderson models. Different curves correspond to different values of the dissipation probability $p$ given in the key. The transport in both settings evolves from diffusive to subdiffusive when one disconnects more and more sites from the baths (i.e., by increasing $p$). The blue dashed lines are the Ohm's law (diffusion with $\beta = 1$). In the insets, we plot the associated exponent $\beta(p)$ (cfr. Eq. (16)) extrapolated at different $p$. The fits are obtained by considering the largest system sizes ($\log L \geq 4$).

for the statistics of the local density of states. For concreteness, we fix $t = 1$, $\Gamma = 1/2$ and $\gamma = 1/4$ for the dephasing protocol, and $t = 1$, $c = 3$ and $\gamma = 1/4$ for the GOE one.

## 4.1 Resistance, Transport and the Phase diagram

The nature of transport is determined by the scaling of the typical resistance with the system size $L$. In particular, a power law dependence

$$R_{\text{typ}} \sim L^{\beta} \tag{16}$$

indicates ballistic transport for $\beta = 0$, while diffusion is signaled by $\beta = 1$, and corresponds to the Ohm's law [79]. Values of $\beta$ that are intermediate between 0 and 1 ($0 < \beta < 1$) are termed superdiffusive while $\beta > 1$ is subdiffusive. The localization limit corresponds to $R \propto \exp(L/\xi_{\text{loc}})$, with $\xi_{\text{loc}}$ the localization length, achieved by a divergent $\beta$.

We computed the stationary current as discussed in Sec. 3.3, and we derived the resistance for the two models (see Eqs. (10) and (11)). In Fig. 4 we display the typical resistance $R_{\text{typ}}$, defined in Eq. (15), of the Anderson model with $W = 3/2$, as a function of system size in double logarithmic scale. We used several values of the probability of decoupling from the local baths, $p$ indicated in the key, in the two protocols, MDB (a) and GOE (b).

In the no-dephasing limit, $p = 1$, the system exhibits Anderson localization for any $W > 0$ and the typical resistance grows exponentially with system size. The divergence of $\beta$ can be observed in the insets where we plot $\beta$ against $p$. In the opposite limit, $p = 0$, in which every site in the chain is subject to dephasing, the system is diffusive $R_{\text{typ}} \propto L$. This limit is shown with a blue broken line in the main plots and as the limit of the $\beta(p)$ curves in the inserts.[3]

Since the fraction $p$ controls how many sites are unaffected by dephasing, as $p$ becomes large there is an increasing number of insulating intervals. As first discussed in Ref. [23], exponentially distributed rare insulating segments with exponentially large resistance produce subdiffusive transport, $R \sim L^{\beta}$ with $\beta > 1$, due to Griffiths effects. The trend of the data shown

---

[3]The apparent super-diffusive behavior at $p = 0$ and small $L$ is due to the fact that when the length of the chain is smaller than the localization length the system behaves as if it were delocalized and hence the transport looks like ballistic.

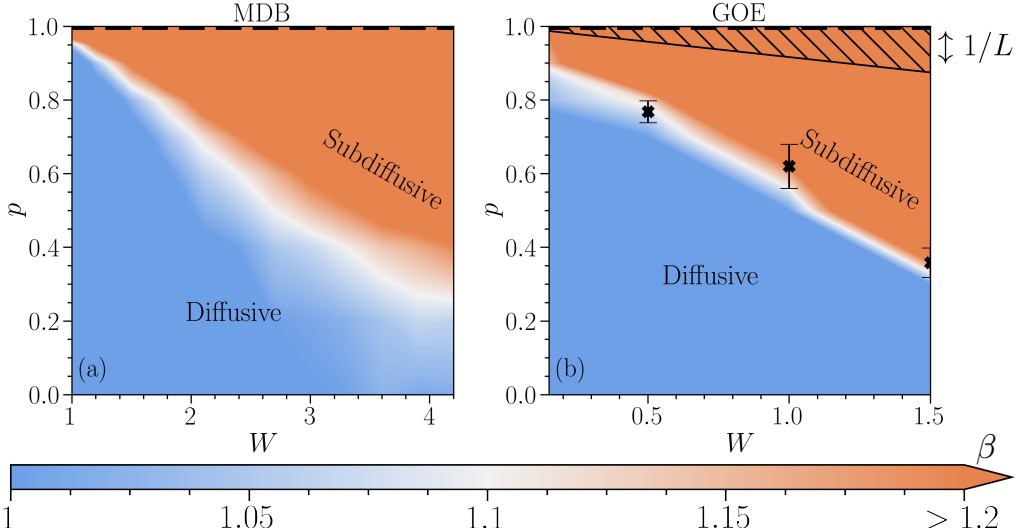

Figure 5: Transport phase diagram of the Anderson model for the MDB (a), and GOE (b) protocols. We obtain the data by extrapolating the system size scaling of the typical resistance $R_{\text{typ}}$ (cfr. Eq. (16)). Specifically, we consider system sizes $L = 8 \div 256$ and measure the exponent $\beta = \log R_{\text{typ}}/\log L$. The white region correspond to points which are diffusive up to two errorbars. The critical line $W_c = W(p_c)$ lies within this area, and signals a transition between a subdiffusive ($\beta > 1$) and a diffusive ($\beta = 1$) behavior. The black cross markers are examples of extrapolation points, with associated error estimations. The black dashed line $p = 1$, corresponds to no dephasing. In this limit, the system is always Anderson localized for any $W > 0$, with a resistance exponentially suppressed in system size (dashed line) The hatched region in the GOE correspond to Anderson localization which only extends in a vanishing region of the phase diagram in the thermodynamic limit where $1 - p = O(1/L)$.

in Fig. 4 complies with these expectations: the curves are ordered from bottom to top for $p$ increasing from $p = 0$ to $p = 0.95$. This is made clear by the double logarithmic scale chosen in the plots.

We note that the GOE bath is less efficient in thermalizing the system compared to the MDB protocol. This is exhibited by the values of the exponent $\beta(p)$ (insets in Fig. 4): the GOE one reaches anomalous (sub) diffusion values at values of $p$ that are smaller than the ones in the MDB protocol. As detailed in Sec. 3.3, this is due to the fact that the band-width of the RRG perturbation is finite and not all the sites in which $\gamma_i > 0$ effectively produce dissipation. The value of $p$ in the GOE setting should thus be "renormalized" to $p_{\text{eff}} \geq p$, as expressed in Eq. (8).

Lastly, we stress that the GOE protocol displays a robust Anderson localization for $1 - p \sim O(1/L)$, as schematically depicted in the figure and already discussed in Ref. [54].

From the analysis of the transport properties discussed so far, in particular the evolution of the exponent $\beta$ in Eq. (16), we can map out a dynamical phase diagram for the Anderson model coupled to the MDB and GOE baths, as a function of $p$ and $W$, which we plot in Fig. 5 (a)-(b). We see that the crossover from diffusive to sub-diffusive transport extends into a crossover line $p_c(W)$, such that for $p < p_c(W)$ the system is diffusive while for $p > p_c(W)$ it is sub-diffusive. The critical dephasing rate decreases with increasing disorder, indicating that a strongly localized Anderson model is more robust to thermal inclusions.

Although, as discussed above, the GOE bath is less efficient to thermalize the samples, the crossover from diffusion to subdiffusion in the two settings is qualitatively very similar.

Finally, we recall that, while the resistance in the MDB setting is computed over all eigenstates, in the GOE case it is computed using only eigenstates around $\omega = 0$. In principle, in

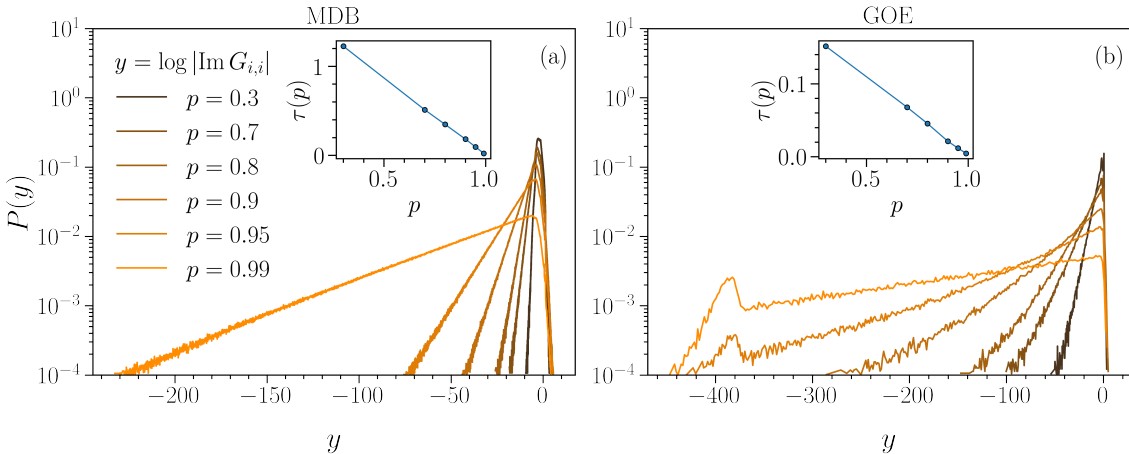

Figure 6: Histograms of the retarded Green functions at $\omega = 0$ of an $L = 512$ Anderson model with $W = 4$, for various values of $p$ in the MDB protocol (a) and the GOE one (b).

the latter case one should also do a finite-size scaling analysis in $M$. In fact, at any finite $M$ we are considering an effective $1d$ disordered system which eventually localizes for $L \gg M$. We can thus expect strong finite size effects and deviations from the power-laws for $L \lesssim M$. Such an analysis would demand a heavy numerical work which goes beyond the scope of this paper.

## 4.2 Statistics of the LDoS

We now discuss how the emergence of sub-diffusive transport and Griffiths affects the spectral properties of the system, as encoded in the imaginary part of the retarded local Green's function, i.e. the local Density of States (LDoS).

The statistics of this quantity allows one to capture the dissipation propagation along the chain. In Ref. [54] it was shown that the sub-diffusive phase of the Anderson model with RRG dissipation is characterized by patchworks of insulating segments where $\text{Im} G_{i,i}(\omega)$ decays exponentially over some length, and rare resonances on which it is of order one. It is therefore interesting to see whether the same physics is captured in the random dephasing Anderson model. For convenience, in the GOE case we fix $\omega = 0$ as a reference energy throughout this subsection. With this choice, in the following we omit the frequency dependence of the LDoS.

In Fig. 6 we plot the histogram of $\log |\text{Im} G_{i,i}|$ in the Anderson model with MDB (a) and GOE (b) baths. Different curves in the two panels correspond to different values of the parameter $p$. Concretely, we fix the system size $L = 512$, and the disorder strength $W = 4$, and we vary $p \in [0,1]$. The histograms display a fat tail towards small values of the argument, $P(y) \sim e^{-\tau(p)|y|}$ with $\tau(p)$ reported in the inserts of the two graphs. In both cases $\tau(p)$ is a decreasing function of $p$ with limit $\tau(p \to 1) \to 0$. However, the $\tau(p)$ takes consistently larger values at $p < 1$ in the MDB protocol than in the GOE one, due to the fact that in the GOE model the relevant probability of un-coupling to the thermal inclusions is $p_{\text{eff}} > p$. We also note that in the GOE case the histogram shows a bump at large negative values of the argument, which becomes more pronounced as $p \to 1$.

Further insights on the shape of LDoS statistics can be obtained from the analysis of $\log |\text{Im} G_{i,i}|$ in single samples. More precisely, we now compare the behavior of the Anderson model with the two ways of including dissipation, by using a single and the same realization of $\{h_i\}$ and $\{\gamma_i\}$. In Fig. 7 (a) and (b) we display the LDoS along the chain. From these single realizations we can appreciate that for small $p$ (i.e., in the diffusive regime) the LDoS fluctuates around values of order one on all the sites of the chain. Instead, for larger $p$ (i.e. in the

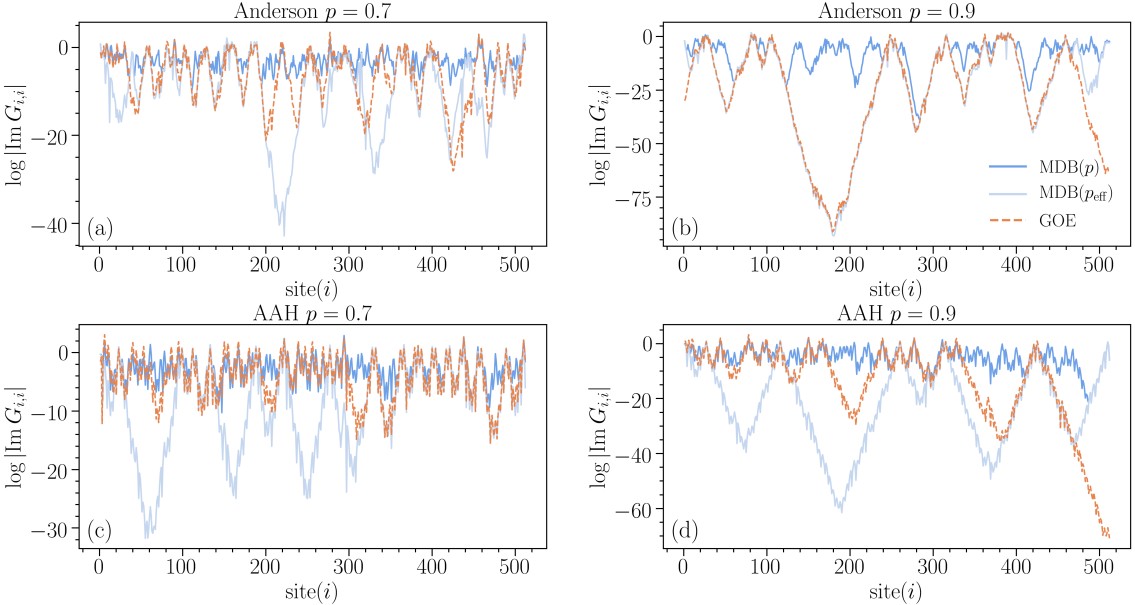

Figure 7: Comparison between the MDB and the GOE protocols for a fixed realization of the couplings $\{h_i, \gamma_i\}$. For the Anderson chain we fix $W = 4$ (a)-(b), while for the AAH we choose $\lambda = 3$ (c)-(d). In the plots we see, once an effective $p_{\text{eff}}$ is considered, the MDB has qualitative features that match those of the GOE protocol.

subdiffusive regime) the range of variation of the LDoS is much broader. Figures 7 (a) and (b) demonstrate that the tails of the distribution in Fig. 6, observed at very small values of the LDoS, originate from a very heterogeneous profile of local dissipation in the system. More precisely, we see that very large insulating segments, within which $\text{Im}\,G_{i,i}$ decreases exponentially over the bare localization length (whose size is exponentially distributed) coexist with a few rare resonances where the bath produces dissipation and $\text{Im}\,G_{i,i}$ is of order 1. In App. C we propose a simple model based on this observation to predict the exponent $\tau(p)$ characterizing the tail of the LDoS distribution. As shown in the inset of Fig. 6, it agrees well with the numerical results. Furthermore, we can understand the origin of the bump in the very tail of the LDoS distribution for the GOE bath in Fig. 6 as being due to the accumulation of samples without resonances (and thus being fully localized).

Finally, the comparison in Fig. 7 of the LDoS profile between the two types of dissipation confirms what already noted, namely that the MDB bath gives rise to deeper minima in the LDoS profile. The GOE model has larger effective probability of un-coupling to the thermal inclusions, $p_{\text{eff}} > p$. If we take into account this difference and renormalize the coupling $\gamma$ to the Markovian dissipation, we obtain an overall better qualitative agreement between the two LDoS profiles at a fixed disorder realization as we show in Fig. 7. We conclude that the analysis of the spectral statistics reveals clear signatures of subdiffusive behavior in the Anderson model with the two kinds of dissipation.

## 5 The André-Aubry-Harper Chain

In full analogy with the Anderson model analysis, we now investigate the transport and dissipative properties of the AAH model with quasi-periodic disorder and random coupling to a MDB or GOE bath. We recall that in the absence of any coupling to the baths the AAH model displays a non-trivial localization transition at $\lambda_c = 2$: for $\lambda < \lambda_c$ the system is ergodic (and

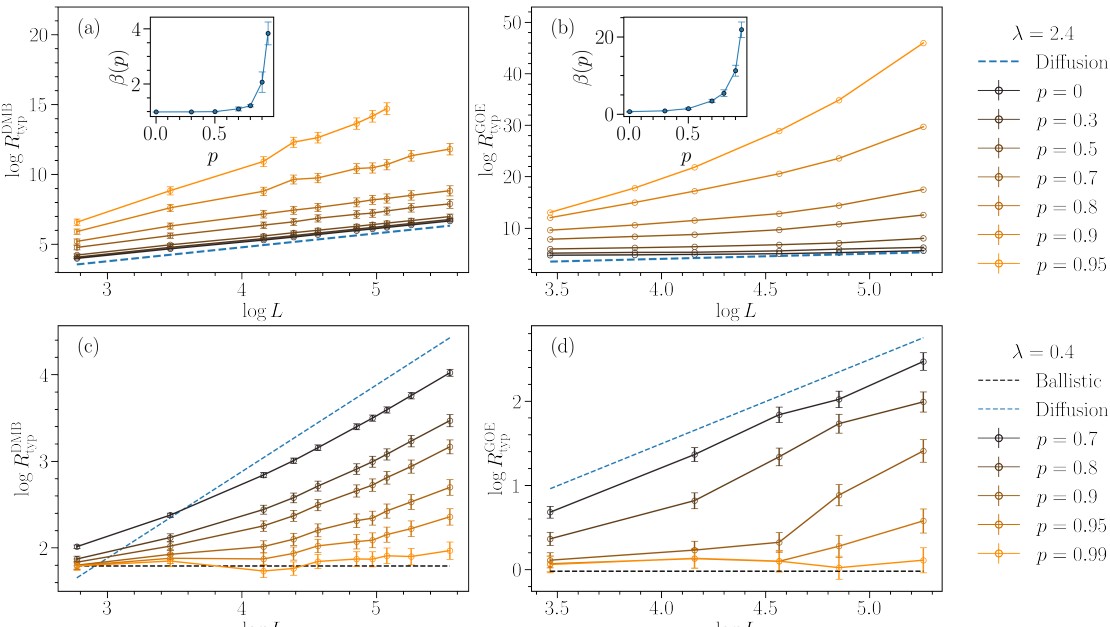

Figure 8: Typical resistance $R_{\text{typ}}$ as a function of system size for the MDB (a) and (c) and GOE (b) and (d) AAH model, for fixed value of the quasi-periodic potential strength $\lambda$, and different values of the dissipation probability $p$. In (a) and (b) $\lambda = 2.4$ and the transport undergoes a transition from diffusive to subdiffusive behavior upon increasing $p$, similarly to what happens in the Anderson model, see Fig. 4. In the insets, we present the exponent $\beta(p)$ (cfr. Eq. (16)) as a function of $p$. In (c) and (d) $\lambda = 0.4$. The asymptotic diffusive limit needs larger systems sizes to establish for increasing $p$ and the long cross-over can be confused with a super-diffusive behavior (see the text for a discussion). The fits are obtained considering the largest system sizes ($\log L \geq 4$).

the transport is ballistic), while for $\lambda > \lambda_c$ the system is Anderson localized. For $p = 1$, i.e. in absence of any bulk coupling to dissipation, the model reduces to the one studied in Ref. [80]. Throughout this section we fix t = 1, $\Gamma = 1/2$ and $\gamma = 1/4$ for the dephasing model, and t = 1, $c = 3$ and $\gamma = 1/4$ for the GOE one.

## 5.1 Resistance, Transport and the Phase Diagram

We start discussing the typical resistance in the steady-state for both the MDB and the GOE AAH models. Our results for $\lambda = 2.4$ are given in Fig 8 (a) and (b), and for $\lambda = 0.4$ in Fig 8 (c) and (d).

First ot all, we see that, in both cases, for sufficiently large system sizes, the typical resistance scales as a power law of system size, i.e. $L^{\beta}$ for any $p < 1$, corresponding to a finite coupling to the baths. The exponent $\beta$ (cfr. Eq. (16)) depends on both the probability $p$ of having a thermal inclusion and the quasi-periodic potential strength $\lambda$ and it is shown in the inset of Fig. 8.

For $\lambda > \lambda_c$, i.e. when the isolated AAH model is in the localized phase, there is a $p_c = p(\lambda_c)$ such that the system is diffusive for $0 \leq p \leq p(\lambda_c)$, and subdiffusive for $p > p(\lambda_c)$, see panels (a) and (b) and the insets where the exponent $\beta$ is seen to deviate from the diffusive limit $\beta = 1$ at $p_c$. For $\lambda \leq \lambda_c$, we see that both models show ballistic transport for $p \to 1$, while for $p < 1$ there is a crossover increasing the system size towards a different scaling behavior which might seem compatible with superdiffusive scaling, $L^{\beta}$ with $\beta < 1$, as recently proposed in Ref. [49, 51] for the interacting and isolated AAH. However, being limited to relatively small system sizes, this analysis alone cannot exclude that the apparent superdiffusive behavior

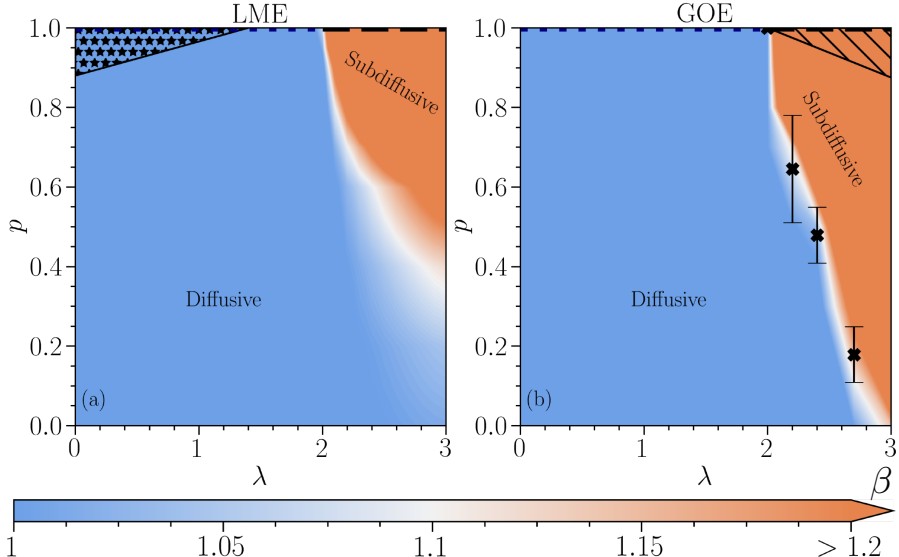

Figure 9: Transport phase diagram of the AAH model coupled to random MDB (a) or GOE (b) baths as a function of the thermal inclusion probability $p$ and the strength of the quasi-periodic potential $\lambda$. At $p = 1$ we see the expected transition from a conducting phase with ballistic transport to a localized one, with exponentially small resistance. As $p$ decreases from one the transport exhibits a transition between a diffusive and a subdiffusive phase. The white region indicates the region where transport is compatible with diffusion up to two error bars. The transition line $\lambda = \lambda_c(p)$ lies in this region. The black cross markers are examples of extrapolation points, with associated error estimations. Close to $p = 1$, in panel (a) the hatched area exhibits a region compatible with superdiffusion, which we cannot fully resolve with our numerical data. Conversely, in panel (b) the hatched area exhibits a region compatible with localization which, similarly, we cannot fully resolve with our numerical precision.

observed at intermediate values of $L$ should eventually disappear in the thermodynamic limit, as recently suggested in Ref. [52] for the interacting many-body AAH. We anticipate that the analysis of the LDoS decline any superdiffusion, supporting a diffusive behavior for any $p < 1$ when $\lambda < \lambda_c$ (in line with Ref. [16, 52]).

Our analysis of the typical resistance is summarized in the transport phase diagram for the AAH model reported in Fig. 9, as a function of the strength of the quasiperiodic potential $\lambda$ and the probability $p$ of having a dissipative coupling. We find that in both dissipative protocols (MDB on the left and GOE on the right panel) there is a crossover line from subdiffusive to diffusive transport at $\lambda_c(p)$ within the localized phase of the model. On the other hand, as already discussed the transport is diffusive for any $p$ for large enough systems and $\lambda < \lambda_c = 2$. Comparing the left panel to the right one we again remark that the GOE bath is less effective in thermalizing the system, with an effective $p_{\text{eff}} > p$.

The emergence of a subdiffusive transport regime in the AAH model under the effect of thermal inclusions, that we found both within the random MDB and the random GOE models, is an intriguing result. To investigate further whether this anomalous transport can be interpreted in terms of Griffiths physics we discuss in the next section the statistics of the LDoS.

## 5.2 Statistics of the LDoS

We conclude the analysis of the AAH model by discussing the statistics of the LDoS across the system evaluated at zero frequency, as done in Sec. 4.2 for the Anderson model. In Fig. 10 we plot the histogram of $\log|\text{Im}\,G_{i,i}|$ for the two types of dissipation considered so far and

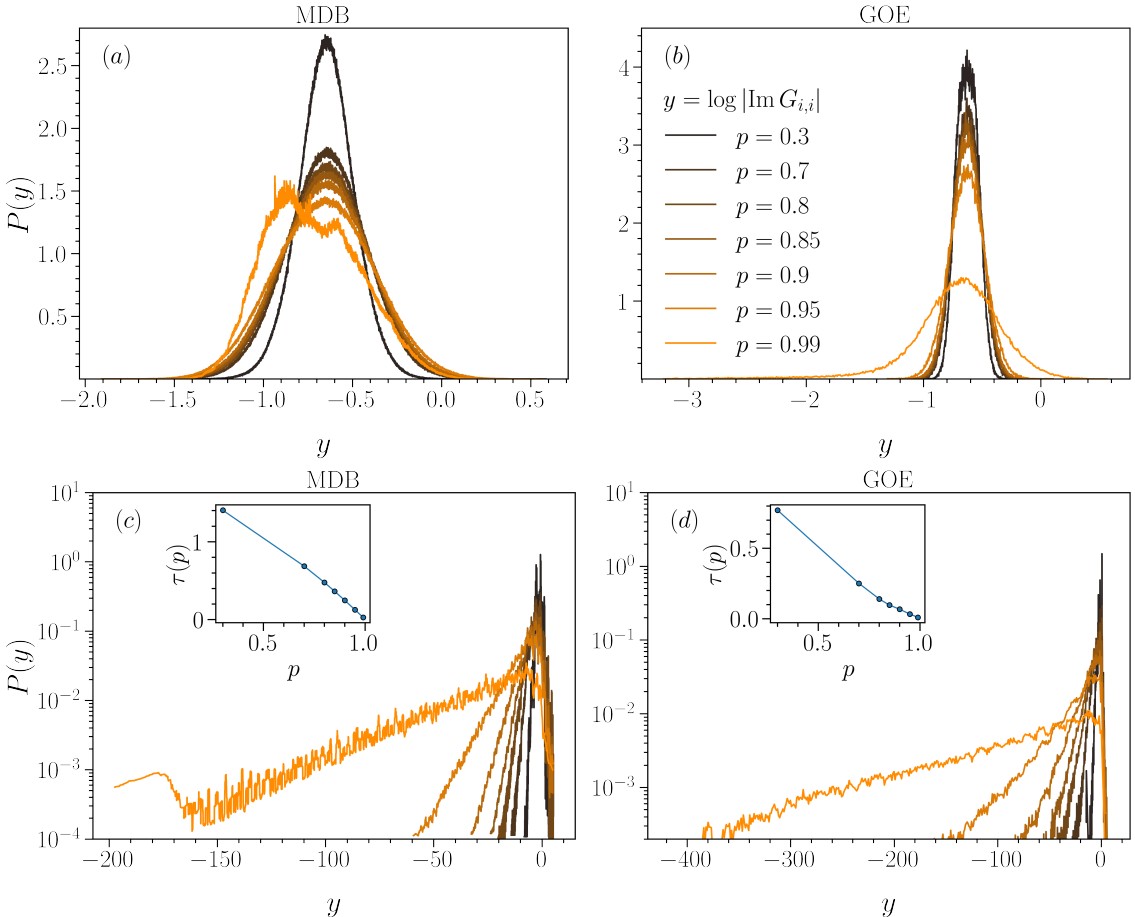

Figure 10: Histograms of the LDoS at $\omega = 0$ for the AAH model with $L = 512$ at $\lambda = 0.4$ (a)-(b) and $\lambda = 3$ (c)-(d) for the random dephasing (top panels) and GOE (bottom panels).

different values of $\lambda$ and $p$. For $\lambda < \lambda_c$ (panels (a) and (b)) we see that the histogram of the LDoS displays a Gaussian profile, modulo some boundary effects for the MDB, with a variance that increases as $p \to 1$. This provide a robust indication that the system is diffusive, as the initial distribution structure of the $h_i$ is washed out by the coupling to the environment. In contrast, this is not the case for $\lambda > \lambda_c$ (see Fig. 10 (c) and (d)) where a power-law behavior describing the distribution at small values of the LDoS emerges as $p$ is increased. As done for the Anderson Model we can extract an exponent describing the power-law decay at large negative arguments, $P(y) \sim e^{-\tau(p)|y|}$ with $\tau(p)$ reported in the inserts of the two graphs. We see that the two exponents are comparable in magnitude, but show an apparent non-linear dependence from $p$. The presence of a broad distribution for the LDoS is consistent with its behavior in single sample realizations that we had plotted in Fig. 7, showing that for $p \to 1$ both models of thermal inclusions lead for $\lambda > \lambda_c$ to a very heterogeneous profile of LDoS with few sites having exponentially small dissipation. The overall picture suggests that the AAH model in its localized phase, when coupled to thermal inclusions, also supports a Griffiths like phenomenology, both for what concerns its transport properties, that turn subdiffusive, as well as for the profile of its effective local dissipation.

# 6 Discussion

In the previous sections we discussed transport and dissipation properties in the AAH and Anderson models in the presence of different kinds of thermal inclusions.

One general outcome of this analysis is that the GOE bath is generically less effective in thermalizing the system and restoring diffusion, and leaves a broader signature of anomalous transport effects in its phase diagram, as compared to the MDB setting, as it appears clearly in the transport phase diagrams of Figs. 5-9. We traced back this difference in the spectral properties of the two dissipative settings. In particular, the Markovian dephasing bath, being essentially broadband, is able to establish resonances for any local energy of the system while the RRG/GOE one with its finite bandwidth requires a resonance condition, see Eq. (8).

An interesting aspect of our results are the similarities that emerged in the properties of the Anderson and AAH models coupled to thermal inclusions, which both show a phenomenology compatible with Griffiths effects, in particular for what concerns the subdiffusive transport and the broad statistics of the LDoS. This is an intriguing and somewhat surprising result at first, given the deterministic nature of the quasiperiodic potential which does not obviously lead to rare region effects. One might wonder how much of these similarities has to do with the nature of the coupling to dissipation, which in both Anderson and AAH models we chose to be random. In this respect it is worth emphasizing that the anomalous properties of the AAH model emerged only for $\lambda > \lambda_c$, i.e. when in absence of any dissipation the system is in a localized phase, while for $\lambda < \lambda_c$ the random coupling to thermal inclusions was shown not to be sufficient by itself to destroy diffusion. As such this result seems to point out that the emergence of subdiffusion and broad dissipation statistics is a robust feature of localized phases coupled to thermal inclusions, irrespectively of the nature of the on-site potential.

Finally, it is interesting to comment on the possible connections between our work and a truly interacting disordered many-body problem in the context of MBL. In the random case it was argued in Refs. [53, 54] that the role of thermal inclusions could mimic the effect of many-body interactions, which are expected to favor local thermalization. As such the Anderson insulator could be a sensible starting point at strong disorder to understand the destruction of a localized phase due to the proliferation of local ergodic regions. In fact the phenomenology found in both Refs. [53, 54] shares many features with the current understanding of the MBL transition. A similar reasoning could be applied to the MBL problem in a quasiperiodic potential. In this respect one could speculate that many-body interactions play a double role, namely to induce local equilibration and at the same time to give rise to a sort of configurational randomness as one could see for example by treating those interactions at the Hartree-Fock level [66, 67]. This reasoning justifies our choice of a random coupling to thermal inclusions also in the AAH case. From this perspective one could consider the results discussed in this work also relevant for the MBL/QP problem in a regime where interactions are weak enough to be treated with HF and the disorder is strong enough to be deep in the localized phase and suggest that some sort of subdiffusive scaling could emerge also in that context.

# 7 Conclusions

In this work we analyzed the effect of local random dissipation on single particle localized phases of matter, as described by the non-interacting Anderson or AAH models. Several reasons, exposed at the beginning of this manuscript, motivated this study.

In the context of quenched random disorder recent works attempted to give a phenological-ical model of the Griffiths phase found near the MBL transition, as arising from coupling to thermal inclusions. Here, we re-examined two of these works, both of which considered An-

derson insulators coupled to different types of dissipative environments, described in terms of random Markovian Dephasing Baths or random GOE baths. In Sec. 4 we provided a thorough comparison of transport and spectral properties of these two settings, thus complementing the existing analysis. As a general outcome we showed that the phenomenology in the two cases is qualitatively similar, with a crossover from sub-diffusive to diffusive transport and the emergence of a broad distribution in the statistics of the local dissipation. Concomitantly, we showed that the GOE bath is typically less effective in thermalizing a localized phase by restoring diffusion, than the Markovian dephasing bath, ultimately due to its finite bandwidth. As a result the properties of an Anderson model coupled to this type of dissipation show anomalous transport and spectral properties in a much broader area of the phase diagram.

A second motivation for this work was to investigate the stability of the localized phase of the quasiperiodic AAH model to thermal inclusions, and to compare its features with the truly random case. In Sec. 5 we extended our analysis to the AAH model coupled to random MDB and GOE dissipation, and we discussed its transport and spectral properties. We provided evidence that even under a quasiperiodic disorder the localized phase is unstable to thermal inclusions towards a phase with Griffiths like phenomenology, including subdiffusive transport and a broad distribution of the local density of states, biased by few rare spots with exponentially small dissipation. Such a phase eventually turns into a conventional diffusive one when the coupling to dissipation becomes more uniform across the chain. On the other hand, we showed that on the delocalized side of the AAH phase diagram this phenomenology is absent and the thermal inclusions lead to diffusive behavior for large enough systems, while finite size samples can display apparent super-diffusive transport. This suggests that the emergence of anomalous transport and spectral properties is a generic feature of localized phases coupled to thermal inclusions, irrespective of the nature of the potential.

In addition to provide two examples of exactly solvable models of disordered systems coupled to random dissipation, both displaying rich physics, our work could be relevant for the MBL problem in the presence of quasiperiodic potentials, as we discuss in Sec. 6. In this respect, it would be interesting to compare more closely the phenomenology that emerges from our toy models with results obtained for the fully interacting case.

## Acknowledgements

**Author contributions**   XT and DB are co-first authors and obtained the results on the Random Markovian bath and the Random Regular Graphs bath, respectively. LFC, MS and MT supervised the project. All authors contributed to the writing of the manuscript.

**Funding information**   MS and XT acknowledge support from the ANR grant "NonEQuMat" (ANR-19-CE47-0001) and computational resources from the Collège de France IPH cluster.

## A   Exact solution for the Markovian Dephasing Bath

We recall that in this setting there is a single chain with sites labeled $i, j$. For convenience, we fix the hopping t.

## A.1 Stationary correlation functions and current

To evaluate the current we only need the evolution of the (instantaneous) spatial correlation matrix

$$C_{i,j}(t) = \text{tr}(\hat{\rho}(t)\hat{d}_i \hat{d}_j^\dagger). \tag{17}$$

In the above equation, we denote time $t$. By differentiating with respect to the time variable $t$ and using the Lindblad equation (5) we find

$$\begin{aligned}
\partial_t C_{i,j}(t) = &-iC_{i-1,j}(t) - iC_{i+1,j}(t) + iC_{i,j+1}(t) + iC_{i,j-1}(t) \\
&- i(h_i - h_j)C_{i,j}(t) + 2\Gamma\delta_{i,L}\delta_{j,L} - 2(1-\delta_{i,j})(\gamma_i + \gamma_j)C_{i,j}(t) \\
&- \Gamma(\delta_{1,j} + \delta_{1,i})C_{i,j}(t) - \Gamma(\delta_{L,j} + \delta_{L,i})C_{i,j}(t).
\end{aligned} \tag{18}$$

Remarkably, despite the fact that dephasing acts as a four body interaction, the equations of motion for the two-point functions are closed and can be solved exactly. We also note that these equations are decoupled from the particle-pair creation/annihilation correlations. In Eq. (18) we implicitly imposed open boundary condition $C_{0,i}(t) = C_{i,L+1}(t) = 0$ for all $i$.

The stationary state is obtained by solving $\partial_t C_{i,j} = 0$ for all $i, j$. For this purpose, it is convenient to rephrase Eq. (18) in matrix form. We introduce the matrix $\mathbb{D}$ capturing the dissipation contributions with elements $D_{i,k} = \delta_{i,k}(\gamma_i + \delta_{i,1}\Gamma + \delta_{i,L}\Gamma)$, and the matrix $\mathbb{P} = \mathbb{P}(\mathbb{C})$, with components $P_{i,k}(t) = 2\delta_{i,k}(\gamma_k C_{k,k}(t) + \Gamma\delta_{i,L}\delta_{k,L}) \equiv p_k(t)\delta_{i,k}$. With these definitions, we recast (18) as

$$\partial_t \mathbb{C} = -i[\mathbb{H}, C] - \{\mathbb{D}, \mathbb{C}\} + \mathbb{P} \equiv \mathbb{T}\mathbb{C} + \mathbb{C}\mathbb{T}^\dagger + \mathbb{P}, \tag{19}$$

where we avoided writing the time dependencies. The infinite time solution is obtained with standard ordinary differential equation tools [53], and reads

$$\mathbb{C}(\infty) = \int_0^\infty dt\, e^{\mathbb{T}t}\, \mathbb{P}(\infty) e^{\mathbb{T}^\dagger t}. \tag{20}$$

We note that this is a self-consistency equation due to the dependency of $\mathbb{P}$ on $\mathbb{C}$. This expression can be further massaged using the spectral decomposition of the matrix $\mathbb{T}$

$$\mathbb{T} = \sum_p \lambda_p |\phi^R(p)\rangle\langle\phi^L(p)|, \qquad \langle\phi^L(q)|\phi^R(p)\rangle = \delta_{p,q}, \tag{21}$$

where with a slight abuse of notation we used the Dirac notation for the $L$-dimensional complex vectors $|\phi^{R/L}(p)\rangle = \sum_i \phi_i^{R/L}(p)|i\rangle$, with components $\phi_i^{R/L}(p)$. Then, from Eq. (21) it follows the series expansion

$$e^{t\mathbb{T}} = \sum_p e^{t\lambda_p} |\phi^R(p)\rangle\langle\phi^L(p)|, \tag{22}$$

which we use in Eq. (20) to derive the self-consistent equation [53, 80]

$$\begin{aligned}
C_{i,j}(\infty) &= 2\Gamma\Theta_{i,j,L} + 2\sum_k \gamma_k \Theta_{i,j,k} C_{k,k}(\infty), \\
\Theta_{i,j,k} &\equiv -\sum_{p,q} \frac{\phi_i^R(p)(\phi_k^L(p))^*(\phi_j^R(q))^*\phi_k^L(q)}{\lambda_p + \lambda_q^*}.
\end{aligned} \tag{23}$$

Lastly, the stationary current is $j_\infty \equiv j_i(\infty) = 2\text{Im}\, C_{i-1,i}(\infty)$. We conclude this subsection by remarking that not all the components of the tensor $\Theta$ are needed to extract the stationary value of the current, which is constant throughout the system.

## A.2 Retarded Green's function

To obtain the retarded Green's function Eq. (12), we need the time dependence of the annihilation operator $\hat{d}(t)$. As presented in Ref. [81, 82], for fermionic fields, this time evolution is generated by the *modified adjoint* Lindbladian

$$\frac{d}{dt}\hat{d}_i(t) = \tilde{\mathcal{L}}^\dagger \hat{d}_i(t), \tag{24}$$

with $\tilde{\mathcal{L}}^\dagger$ defined as

$$\tilde{\mathcal{L}}^\dagger(\circ) \equiv i[\hat{H}, \circ] + \mathcal{D}_d[\circ] + \tilde{\mathcal{D}}_{\mathrm{bnd,l}}[\circ] + \tilde{\mathcal{D}}_{\mathrm{bnd,r}}[\circ], \tag{25}$$

$$\tilde{\mathcal{D}}_{\mathrm{bnd,l}}[\circ] = \Gamma\left(2\eta\hat{d}_1 \circ \hat{d}_1^\dagger - \{\hat{d}_1\hat{d}_1^\dagger, \circ\}\right), \tag{26}$$

$$\tilde{\mathcal{D}}_{\mathrm{bnd,r}}[\circ] = \Gamma\left(2\eta\hat{d}_L^\dagger \circ \hat{d}_L - \{\hat{d}_L^\dagger\hat{d}_L, \circ\}\right). \tag{27}$$

In the above equation $\hat{H}$ and $\hat{\mathcal{D}}_{\mathrm{d}}$ are as in Eq. (6), while the phase $\eta = -1$ fully captures the fermionic nature of the degrees of freedom. The final expression for the evolution equation of the retarded Green functions is

$$\frac{d}{dt}G_{i,j}(t) = -i\delta_{i,j}\delta(t) - i\sum_k H_{i,k}G_{k,j} - \gamma G_{i,j}(t) - \Gamma(\delta_{1,i} + \delta_{L,i})G_{i,j}(t). \tag{28}$$

Recasting it in matrix form, and using the previously defined $\mathbb{T} = -i\mathbb{H} - \mathbb{D}$, we find

$$\frac{d}{dt}\mathbb{G}(t) = -i\delta(t)\mathbb{1} + \mathbb{T}\mathbb{G}(t). \tag{29}$$

Fourier transforming both sides, we have

$$\mathbb{G}(\omega) = (\omega\mathbb{1} - i\mathbb{T})^{-1}. \tag{30}$$

In the specific case of homogeneous systems, a closed expression for the matrix elements in Eq. (30) can be obtained [69, 83].

## B The Cavity Method

The diagonal elements of the Green's function of the model (7) can be in principle computed exactly using a Transfer Matrix approach. This can be done, for instance, by direct Gaussian integration. The starting point is the formal representation of the Green's function as an integral over the bosonic variables $\{\phi_{i,n}, \phi_{i,n}^\star\}$ defined on each node $n = 1, \dots, M$ belonging to the $i$-th layer:

$$G_{i',n';j',m'} = \frac{\int \mathcal{D}\phi\mathcal{D}\phi^\star \, \phi_{i',n'}^\star \phi_{j',m'} \, e^{-\sum_{i,j=1}^L \sum_{n,m=1}^M \phi_{i,n}^\star (H-zI)_{i,n;j,m}\phi_{j,m}}}{\int \mathcal{D}\phi\mathcal{D}\phi^\star \, e^{-\sum_{i,j=1}^L \sum_{n,m=1}^M \phi_{i,n}^\star (H-zI)_{i,n;j,m}\phi_{j,m}}}.$$

Let us now imagine that we have progressively integrated out all the bosonic variables on all the sites of the layers $i < j \leq L$ starting from the rightmost one. Since the action above is quadratic, the result of this integartion will also be quadradic and can be formally encoded in a Gaussian weight describing the measure over the bosonic variables of the $i$-th layer in absence of all the layers on the right:

$$P_i^{(r)}[\{\phi_{i,n}, \phi_{i,n}^\star\}] \propto e^{-\sum_{n,m} \phi_{i,n}^\star \left[G_i^{(r)}\right]_{nm}^{-1} \phi_{i,m}},$$

where $G_i^{(r)}$ is an $M \times M$ complex symmetric matrix (and similarly for the measure in absence of the left neighboring layer). By Gaussian integration, it is straightforward to show that the recursion relations for the left and right elements of the measure read:

$$
\begin{aligned}
\left[G_i^{(r)}\right]_{nm}^{-1} &= (-h_i - z)\delta_{n,m} - \gamma_i^2 \mathcal{C}_{nm}^{(i)} - t^2 G_{i-1,nm}^{(r)}, \\
\left[G_i^{(l)}\right]_{nm}^{-1} &= (-h_i - z)\delta_{n,m} - \gamma_i^2 \mathcal{C}_{nm}^{(i)} - t^2 G_{i+1,nm}^{(l)}.
\end{aligned}
\tag{31}
$$

Here, $\mathcal{C}_{nm}^{(i)}$ is the connectivity matrix of the RRG on the $i$-th layer whose elements are equal to 1 if $n$ and $m$ are connected by the RRG and zero otherwise, and $\gamma_i = \gamma$ with probability $1 - p$ and zero with probability $p$. Considering an open system the boundary conditions on $i = 1$ and $i = L$ are

$$
\begin{aligned}
\left[G_1^{(r)}\right]_{nm}^{-1} &= (-h_1 - z)\delta_{n,m} - \gamma_1^2 \mathcal{C}_{nm}^{(1)}, \\
\left[G_L^{(l)}\right]_{nm}^{-1} &= (-h_L - z)\delta_{n,m} - \gamma_L^2 \mathcal{C}_{nm}^{(L)}.
\end{aligned}
\tag{32}
$$

The first two equations for the Green's functions can be easily solved by inverting the matrices on the right hand side by lower-upper (LU) decomposition [78] on each layer. Note that these equations are exact but can only be solved for finite $M$.

However, in the $M \to \infty$ limit a drastic simplification arises. In this limit, the typical length of the loops in the transverse planes diverges and the Gaussian measure on the $c$ neighbors of a given site within a given layer factorizes in absence of the central site, since these $c$ neighbors belong to disconnected branches of the graph and are uncorrelated. Moreover, as explained in the main text, since the local potential is the same on all the vertices of the transverse planes, in the $M \to \infty$ limit they all become statistically equivalent and a translational invariance within each plane is restored. Equations (31) and (32) are thus drastically simplified. Following Ref. [54], one can show that:

$$
\left[G_i^{(r)}\right]^{-1} = h_i - z - t^2 G_i^{(r)} - c\gamma_i^2 G_i^{(v)},
\tag{33}
$$

$$
\left[G_i^{(l)}\right]^{-1} = h_i - z - t^2 G_i^{(l)} - c\gamma_i^2 G_i^{(v)},
\tag{34}
$$

with $c$ the uniform connectivity of the RRG on the planes, and

$$
\left[G_i^{(v)}\right]^{-1} = h_i - z - t^2 G_{i+1}^{(r)} - t^2 G_{i-1}^{(l)} - (c-1)\gamma_i^2 G_i^{(v)},
\tag{35}
$$

where $G_i^{(r,l,v)}$ are the diagonal elements of the Green's function on any site of the $i$-th layer in absence of its right neighbor, its left neighbor, and one of its $c$ neighbors in the transverse plane, respectively. Once these so-called "cavity" Green's functions are known on each layer, one can finally compute the diagonal elements of the Green's function of the original problem on layer $i$ (see Ref. [54] for more details):

$$
\mathcal{G}(z) = (\hat{H}_{\text{GOE}} - z\hat{\mathbb{1}})^{-1},
\tag{36}
$$

via

$$
[\mathcal{G}_i(z)]^{-1} = h_i - z - t^2 G_{i+1}^{(r)} - t^2 G_{i-1}^{(l)} - c\gamma_i^2 G_i^{(v)}.
\tag{37}
$$

We insist upon the fact that there is no dependence on the in-layer indices $n, m$ within this approximation. Besides, focusing on the limiting case of independent layers ($t = 0$) we can exactly obtain the density of states in each layer. As a matter of fact for this case we can solve Eq. (35) and obtain

$$
G_i^{(v)} = \frac{h_i - z \pm \sqrt{(h_i - z)^2 - 4(c-1)\gamma_i^2}}{2\gamma_i^2 (c-1)}.
\tag{38}
$$

Then with Eq. (37) we have for the Local Density of States (LDoS) that

$$\mathrm{Im}\mathcal{G}_i(z) \propto \mathrm{Im}G_i^{(v)}. \tag{39}$$

Therefore, on a given layer $i$ the LDoS can be non-zero only if $|h_i - z| \leq 2\sqrt{c-1}\gamma_i$. More generally, and as performed in [54], with Eqs. (33), (34), and (35) we can derive the LDoS in the limit $M \to +\infty$ for any set of coupling parameters.

However, this approach is not suited to compute the resistance $R(\omega)$ as a large number of paths (all those containing loops) connecting the first and last layer are neglected. Indeed, with this approximation, the resistance is always overestimated and we always obtain a localized regime, i.e $R(\omega)$ scales as $R(\omega) \sim e^{\alpha L}$ ($\alpha > 0$).

One way to overcome this problem and compute the resistance is to go back to the finite $M$ case and solve recursively the set of exact Eqs. (31). With their solution one can then derive the conductivity $\sigma(\omega)$ at energy $\omega$ following the approach developed in [74] by Fisher and Lee. The set-up consists in connecting two reservoirs – also called leads – to the layers $i = 1$ and $i = L$ and noticing that $\sigma(\omega)$ is simply given by the matrix elements of the resolvent – up to a proportionality factor – according to

$$
\begin{aligned}
\sigma(\omega) &= \lim_{\eta \to 0^+} \frac{1}{M} \left| \sum_{n_1, n_L = 1}^{M} \langle 0 | d_{L, n_L} \mathcal{G}(z) d_{1, n_1}^\dagger | 0 \rangle \right|^2 \\
&= \lim_{\eta \to 0^+} \frac{1}{M} \left| \mathrm{t}^{L-1} \sum_{\{n_k\}_{k \in [\![1, L+1]\!]}} \left[ G_L^{(l)} \right]_{n_L, n_{L+1}} \prod_{i=1}^{L-1} \left[ G_i^{(r)} \right]_{n_i, n_{i+1}} \right|^2.
\end{aligned} \tag{40}
$$

It is important to note that the reservoirs modify the boundary conditions of Eq. (31). At the edges, $i = 1$ and $i = L$,

$$
\begin{aligned}
\left[ G_i^{(l)} \right]_{nq}^{-1} &= [-h_i - z - \Sigma(\omega)] \delta_{n,q} - \gamma_i \mathcal{C}_{nq}^{(i)} - \mathrm{t}^2 G_{i+1, nq}^{(l)}, \tag{41} \\
\left[ G_i^{(r)} \right]_{nq}^{-1} &= [-h_i - z - \Sigma(\omega)] \delta_{n,q} - \gamma_i \mathcal{C}_{nq}^{(i)} - \mathrm{t}^2 G_{i-1, nq}^{(r)},
\end{aligned}
$$

where $\Sigma(\omega)$ is the self-energy of the leads verifying

$$\Sigma = -E + i\frac{\sqrt{4\mathrm{t}^2 - E^2}}{2E}. \tag{42}$$

The resistance is then the inverse of the conductivity, $R^{\mathrm{GOE}}(\omega) = 1/\sigma(\omega)$.

Alternatively, It is also possible to derive the current $j_\infty^{\mathrm{GOE}}(\omega)$ by adding source terms at the boundaries $i = 1$ and $i = L$ in the cavity measures $P_i^{(r)}[\{\phi_{i,n}, \phi_{i,n}^\star\}]$ and $P_i^{(l)}[\{\phi_{i,n}, \phi_{i,n}^\star\}]$ (for more details, see [84]). This approach is equivalent to the previous one and gives – up to a proportionality factor – the same resistance $R^{\mathrm{GOE}}(\omega)$.

## C  Predicting the tail in the LDoS

Focusing on Fig. 7 we can note that the LDoS profiles are characterized by a handful of resonances ($\mathrm{Im}G_{i,i} = \mathcal{O}(1)$) interspersed by regions where the density of states decays exponentially. In these isolating regions the LDoS scales as $\mathrm{Im}G_{i,i} \sim e^{-|i-i_o|/\xi_{\mathrm{loc}}}$ with $i_o$ the position of the closest resonance and $\xi_{\mathrm{loc}}$ the bare localization length. In particular, within a localized region of length $l$ the LDoS appears to be distributed uniformly with values ranging from

around $e^{-l/2\xi_{\text{loc}}}$ (the minimum value of the density being attained roughly at the middle of the segment) to $\mathcal{O}(1)$. Following the analysis in [53] (in particular the third section) the total number of layers with $\log|\text{Im}(G_{i,i})|$ equal to $-\alpha$ (up to sub-leading terms) is

$$\Omega\big[\log|\text{Im}(G_{i,i})| = -\alpha\big] \approx \sum_{l \geq 2\xi_{\text{loc}}\alpha} 2n(l), \tag{43}$$

where $n(l)$ is the number of isolating segments of length $l$ in the system. Equation (43) simply translates the fact that only isolating segments longer than $2\xi_{\text{loc}}\alpha$ contain layers where $\log|\text{Im}(G_{i,i})| = -\alpha$: these layers being approximately two in number, one when the LDoS decreases exponentially with $i$, and one when the LDoS increases exponentially with $i$. Then, the probability distribution is given by

$$P\big[\log|\text{Im}(G_{i,i})| = -\alpha\big] \approx \frac{\Omega\big[\log|\text{Im}(G_{i,i})| = -\alpha\big]}{L} \approx \frac{2}{L}\sum_{l \geq 2\xi_{\text{loc}}\alpha} n(l). \tag{44}$$

In fact, if we consider that each layer in the system has a probability $p$ to form a local resonance in the LDoS ($\text{Im}G_{i,i} = \mathcal{O}(1)$) it can be proven that $n(l)$ is a random variable following the Poisson distribution [53]

$$P[n(l)] = \frac{\mu(l)^{n(l)}}{n(l)!}e^{\mu(l)}, \quad \text{with} \quad \mu(l) = L(1-p)p^l. \tag{45}$$

Averaging the probability density $P\big[\log|\text{Im}(G_{i,i})| = -\alpha\big]$ over the distribution of $n(l)$ then yields

$$
\begin{aligned}
\big\langle P\big[\log|\text{Im}(G_{i,i})| = -\alpha\big]\big\rangle &\approx 2\sum_{l \geq 2\xi_{\text{loc}}\alpha}(1-p)^2 p^l \\
&\approx 2\sum_{k \geq 0}(1-p)^2 p^{k+2\xi_{\text{loc}}\alpha} \\
&\approx 2(1-p)p^{2\xi_{\text{loc}}\alpha},
\end{aligned}
\tag{46}
$$

where $\langle\circ\rangle$ indicates the average over all variables $\{n(l)\}_{l \in \mathbb{N}}$. For $p$ close to one this simple approach predicts a tail $\tau(p)$ in the probability distribution

$$\big\langle P\big[\log|\text{Im}(G_{i,i})| = -\alpha\big]\big\rangle \sim e^{-\alpha\tau(p)}, \qquad \text{with} \qquad \tau(p) = -2\xi_{\text{loc}}\log(p). \tag{47}$$

When compared to our numerical results we observe a good agreement with this prediction, see the inset in Fig. 6.

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
