# Peer review of "Destruction of Localization by Thermal Inclusions: Anomalous Transport and Griffiths Effects in the Anderson and André-Aubry-Harper Models"

_SciPost Physics, doi:SciPost Phys. 12, 189 (2022)_

## Round 2 · Referee Report · Anonymous (Referee 1) · 2022-1-25

Strengths

1-Clearly written with good quality figures.
2-Rather complete study of two models of thermal inclusion in localised systems, comparing random disorder with quasi-periodic potentials

Weaknesses

1-Some of the appendices could be more self-contained.

Report

This is a careful study of two models of thermal inclusions in localised systems. One where the thermal regions are modelled by local dephasing and second where they are modelled by coupling to random regular graphs. Both models have been studied in detail in the case of random disorder (Refs. 58 and 59). The main contribution of this work is to extend these studies to quasi-periodic potentials, namely the André-Aubry-Harper chain, and compare and contrast the two models of thermal inclusion in the quasi-periodic case to the random case. The main conclusion of the work is that these two cases behave qualitatively the same (apart from the fact that the André-Aubry-Harper chain has a localisation transition in the absence of dephasing).

As both models have been studied in detail, this work may read as slightly incremental. There is new data provided for the random case, complementing the already published data, but not really providing any new insight. However, even if the behaviour of the quasi-periodic case is qualitatively the same, I do think the paper raises some questions and can likely inspire some follow-up work, and therefore I believe it likely satisfies the third expected acceptance criteria of SciPost Physics. The authors may need to comment on wether their work satisfies the 5th general acceptance criteria, requesting that all reproducing-enabling resources be provided.

I have some comments on the manuscript below. My main question after reading the paper is whether one could have expected anything different for the results. I find it interesting to compare potential Griffiths effects in quasi-periodic potentials but to what extend is that really being done here, since the thermal inclusions are still being modelled by randomly located dephasers or random regular graphs. The authors briefly mention this question but I did not quite feel that question was answered. It would be useful to have a bit more extended discussion, perhaps backed up by some data if possible, on this point, if something more can be said about it.

Requested changes

1-In the main text, when discussing the energy dependence in the GOE case, it would be useful to explain the mode of the leads and what energy they are talking about here.
2-After Eq. (14): "Thanks to the translational invariance within the transverse planes." Not sure what they mean here. Aren't these random regular graphs.
3-Figure 4. Can the other provide some more details of the fitting to obtain $\beta(p)$. What are the error bars? Does $\beta(p)$ flow with system size $L$? The authors mention that the super-diffusion can be, or is, only apparent but will go to diffusion at large system sizes. Can they rule out that similar flow of the exponent does not happen for the sub-diffusive case and that it will flow to diffusion at very large systems? I am here interested mainly in wether something can be said with the available data, not some hypothetical even larger system sizes than they can reach.
4-Is there a reason to not include the curves for $p=1$ in, for example, Fig. 4. It could be useful to have that limit as well.
5-Figure 5. What is being plotted in the color scale and what is the scale. The caption seems to suggest that something binary is being plotted, but the white region would suggest otherwise.
6-In the text explaining Figure 7, the other discuss what happens at small $p$, but no data is provided in the case of small $p$. Would it be useful to include that data? Also the caption to the figure simply says "Comparison" but not comparison of what. Finally: the authors discuss how the two models are qualitatively the same after introducing a $p_\text{eff}$, yet in some cases a peak in one model gives a valley in the other. Not sure how seriously one should take this comparison.
7-Maybe add also $\gamma_i$ to Figure 7?
8-Since the first two authors are co-first authors, it would maybe make sense to mark that somehow also in the author list , not only in the acknowledgment.
9-Appendix A: The authors fix $J=1$, likely this should be $t=1$.
10-Notation in Eq. (19). Doesn't $P$ depend on $C$?
11-Appendix B could be made a bit more self contained by a few more sentences of introduction. What are the boson variables, what is the starting point, etc. The could explain more clearly what loops they are referring to when they neglect all loops. And $G_i^{v}$ is not defined. This notation is used in their earlier work but not defined here.
12-There are several references that are not complete. This seems to be mainly arXiv references that are missing the arXiv numbers.

Typos and minor things
1-Throughout analysis should be thorough analysis.
2-Figure 1: Sum in $\mathcal{D}_d$ should be over $i$ instead of $I$.
3-Couples of sites $\rightarrow$ pairs of sites
4-In Eq. (8). $k$ is not defined. Maybe it was supposed to be $c-1$?
5-Just before section 5.1, $t$t=1 $\rightarrow$ $t=1$
6-Page 17. A sentence starts with "On the other,".
7-Eq. (22): should $\psi \rightarrow \phi$
8-Eq. (28): should $\delta_{i,j}(t) \rightarrow \delta_{i,j}\delta(t)$.
9-Is there an extra minus sign in Eq. (48)?

---

## Round 2 · Referee Report · Anonymous (Referee 2) · 2022-1-31

Report

This manuscript comprises a detailed numerical analysis of two toy models conceived to study the breakdown of localization in disordered non-interacting chains. Both of these models introduce locally dissipative effects designed to mimic the effect of delocalised regions. The authors provide an exhaustive comparison of the two models, providing convincing evidence that they may both be understood in terms of Griffiths physics. Specifically, that their transport properties are dictated by rare but large localised regions in which there are no dephasing effects. This is shown through a detailed analysis of both typical quantities and their distributions.

The manuscript is well written, and, after accounting for the requested changes below, I would recommend it be published in Sci-post, for which it is suitable.

Requested changes

--- Recent work has developed the idea (see Refs [53], [54]) that the thermodynamic MBL transition and the finite size MBL “transition” may be driven by distinctly different physics. With [54] noting indeed that “the MBL phase transition actually occurs … far from the numerically-accessible crossover between the finite-size MBL regime and thermalization, reinforc[ing] the idea that the physics of this crossover is likely quite different from that of the ultimate phase transition. This suggests that this crossover should probably be studied as a distinct phenomenon from the MBL phase transition.” I would encourage the present authors to make clearer in their introduction which regime the authors are focussing on: the physics of the thermodynamics transition, or the physics of the numerically accessible transition (I assume the former, as it is in this regime in which spatial variation and Griffiths effects are believed to be important), and to make sure their citations are consistent with this.

--- In Eq 3 I think $k = 1 – c$, however I think $k$ was not introduced in the text.

--- In Figs 5 and 9 the procedure used to make the density plot was unclear. Some explanation and a color bar indicating precisely how I should interpret intermediate colors would seem necessary.

--- Regarding the comment: “As such this result seems to point out that the emergence of subdiffusion and broad dissipation statistics is a robust feature of localized phases coupled to thermal inclusions, irrespectively of the nature of the on-site potential.” It seems important to qualify this with a comment that it applies in the case where the dephasing dynamics is *assumed* to apply to a randomly distributed set of sites. I am aware that the authors make a comment that configurational disorder may potentially lead to this situation in quasiperiodically modulated many body system, however this later comment is somewhat conjectural, whereas the comment above is much more sweeping.

---

## Round 3 · Author Response

Referee #1

Report

This manuscript comprises a detailed numerical analysis of two toy models conceived to study the breakdown of localization in disordered non-interacting chains. Both of these models introduce locally dissipative effects designed to mimic the effect of delocalised regions. The authors provide an exhaustive comparison of the two models, providing convincing evidence that they may both be understood in terms of Griffiths physics. Specifically, that their transport properties are dictated by rare but large localised regions in which there are no dephasing effects. This is shown through a detailed analysis of both typical quantities and their distributions. The manuscript is well written, and, after accounting for the requested changes below, I would recommend it be published in Sci-post, for which it is suitable.

We warmly thank the referee for the positive assessment of our paper.

Requested changes 1.

Recent work has developed the idea (see Refs [53], [54]) that the thermodynamic MBL transition and the finite size MBL “transition” may be driven by distinctly different physics. With [54] noting indeed that “the MBL phase transition actually occurs … far from the numerically-accessible crossover between the finite-size MBL regime and thermalization, reinforc[ing] the idea that the physics of this crossover is likely quite different from that of the ultimate phase transition. This suggests that this crossover should probably be studied as a distinct phenomenon from the MBL phase transition.” I would encourage the present authors to make clearer in their introduction which regime the authors are focussing on: the physics of the thermodynamics transition, or the physics of the numerically accessible transition (I assume the former, as it is in this regime in which spatial variation and Griffiths effects are believed to be important), and to make sure their citations are consistent with this.

We thank the referee for this remark. We have added a whole new paragraph in the introduction to discuss this important point. In this paper we focus on the properties of the sub-diffusive regime which precedes the MBL transition which, as recently suggested in the works mentioned by the referee, may be driven by distinctly different physical mechanisms compared to the MBL transition itself. In this regard, the advantage of working with simplified toy models is twofold. On one hand, they allow one to inspect the physics on much larger scales (on which Griffiths effects are indeed believed to be important) compared to truly interacting models. On the other hand, they allow one to study the sub-diffusive regime disentangling it from the MBL transition.

  1. In Eq 3 I think k=1–c, however I think k was not introduced in the text.

Yes, the referee is right. Whe have replaced $k->c-1$.

3.

In Figs 5 and 9 the procedure used to make the density plot was unclear. Some explanation and a color bar indicating precisely how I should interpret intermediate colors would seem necessary.

We have added a bar with the color scale and extended the captions of both figures to provide a clearer explanation.

Regarding the comment: “As such this result seems to point out that the emergence of subdiffusion and broad dissipation statistics is a robust feature of localized phases coupled to thermal inclusions, irrespectively of the nature of the on-site potential.” It seems important to qualify this with a comment that it applies in the case where the dephasing dynamics is assumed to apply to a randomly distributed set of sites. I am aware that the authors make a comment that configurational disorder may potentially lead to this situation in quasiperiodically modulated many body system, however this later comment is somewhat conjectural, whereas the comment above is much more sweeping.

The referee is right. A similar remark was made by referee #2. We have added a long new paragraph at the end of Sec. II to discuss this important issue in more detail. We agree with the referee that for a truly many-body model in a quasiperiodic potential the spatial structure of the thermal inclusions is not random and is arguably correlated with the potential itself. However, predicting the position of these thermal spots is a formidable task, which is essentially equivalent to solving the many-body problem. For this reason in the toy models we work with, the dephasing dynamics is assumed to apply on a randomly distributed set of sites, independently of the local potential, both in the Anderson and in the AAH case. One could then argue that, since in our simplified setting the insulating regions and the thermal inclusions are essentially put by hand at random, one cannot draw any conclusion on the universality of the Griffiths picture. Yet, on the one hand one could speculate that many-body interactions, together with the randomness of the initial configuration, give rise to some sort of configurational disorder, as one could see for example by treating those interactions at the Hartree-Fock level (see e.g. Refs.~[66, 67]). On the other hand, we would like to point out that the fact that the Anderson model with uncorrelated disorder and the AAH model in the localized regime respond in a very similar way when coupled to a thermalizing system is still interesting and informative. In fact, as explained in the paper, in the Anderson model the formation of rare resonances is a quite “simple” local process: resonances are formed on the sites on which the local disorder is small and the dephasing dynamics is active. Instead, the formation of resonances in the AAH case cannot be predicted easily and is likely to involve more complex and non-local processes. Still, the phenomenology of the two models when coupled to the thermal inclusions is essentially the same, for both kinds of dissipative environments that we consider. We believe that this gives some strong indication on the fact that the subdiffusive regime is a very robust feature which appears whenever Anderson localized edgestates are perturbed by thermal inclusions, independently of the details of the microscopic modelization of the bath.

Referee #2

Strengths 1-Clearly written with good quality figures. 2-Rather complete study of two models of thermal inclusion in localised systems, comparing random disorder with quasi-periodic potentials

We warmly thank the referee for the positive assessment of our work.

Weaknesses 1-Some of the appendices could be more self-contained.

We have modified the appendices (especially App. B) to make them more self-contained (see below).

Report

I have some comments on the manuscript below. My main question after reading the paper is whether one could have expected anything different for the results. I find it interesting to compare potential Griffiths effects in quasi-periodic potentials but to what extend is that really being done here, since the thermal inclusions are still being modelled by randomly located dephasers or random regular graphs. The authors briefly mention this question but I did not quite feel that question was answered. It would be useful to have a bit more extended discussion, perhaps backed up by some data if possible, on this point, if something more can be said about
it.

We agree with the referee on this point (which has also been raised by referee #1).We have added a long new paragraph at the end of Sec. II to discuss this important issue in more detail. We agree with the referee that for a truly many-body model in a quasiperiodic potential the spatial structure of the thermal inclusions is not random and is arguably correlated with the potential itself. However, predicting the position of these thermal spots is a formidable task, which is essentially equivalent to solving the many-body problem. For this reason in the toy models we work with, the dephasing dynamics is assumed to apply on a randomly distributed set of sites, independently of the local potential, both in the Anderson and in the AAH case. One could then argue that, since in our simplified setting the insulating regions and the thermal inclusions are essentially put by hand at random, one cannot draw any conclusion on the universality of the Griffiths picture. Yet, on the one hand one could speculate that many-body interactions, together with the randomness of the initial configuration, give rise to some sort of configurational disorder, as one could see for example by treating those interactions at the Hartree-Fock level (see e.g. Refs.~[66, 67]). On the other hand, we would like to point out that the fact that the Anderson model with uncorrelated disorder and the AAH model in the localized regime respond in a very similar way when coupled to a thermalizing system is still interesting and informative. In fact, as explained in the paper, in the Anderson model the formation of rare resonances is a quite “simple” local process: resonances are formed on the sites on which the local disorder is small and the dephasing dynamics is active. Instead the formation of resonances in the AAH case cannot be predicted easily and is likely to involve more complex and non-local processes. Still, the phenomenology of the two models when coupled to the thermal inclusions is essentially the same, for both kinds of dissipative environments that we consider. We believe that this gives some strong indication on the fact that the subdiffusive regime is a very robust feature which appears whenever Anderson localized edgestates are perturbed by thermal inclusions, independently on the details of the microscopic modelization of the bath.

Requested changes

1-In the main text, when discussing the energy dependence in the GOE case, it would be useful to explain the mode of the leads and what energy they are talking about here.

We have added a sentence after Eq. (11) to make clear that throughout the paper we only measure the conductance at \omega=0, around the middle of the single-particle spectrum.

2-After Eq. (14): "Thanks to the translational invariance within the transverse planes." Not sure what they mean here. Aren't these random regular graphs.

In fact the typical length of the loops in the transverse planes diverges as log M. Therefore, since the local potential is the same on each node of the transverse layer, in the thermodynamic limit they all become statistically equivalent and the translational invariance in the transverse direction is recovered. We have added a sentence after Eq. (14) to explain this point.

3-Figure 4. Can the other provide some more details of the fitting to obtain β(p). What are the error bars? Does β(p) flow with system
size L? The authors mention that the super-diffusion can be, or is, only apparent but will go to diffusion at large system sizes. Can they
rule out that similar flow of the exponent does not happen for the sub-diffusive case and that it will flow to diffusion at very large
systems? I am here interested mainly in wether something can be said with the available data, not some hypothetical even larger
system sizes than they can reach.

The fits are obtained by considering the largest system sizes (log L ≥ 4). We have added the error bars in Figure 4 and explained how we did the fits in the caption. We agree with the referee on the fact that β(p) has finite size corrections. The flowing area corresponds to the white regions in Fig. 5 and Fig. 9, where our numerics does not allow precise allocation of the transition. However, we would like to point out that, while for the resistivity, for technical reasons we can only access relatively small sizes and one might still have doubts on what happens in the infinite size limit, when we study the statistics of the LDoS we can reach much larger sizes and observe that the signatures of sub-diffusion persist at large L. In fact the qualitative pictures of few rare resonances separated by large insulating regions highlighted in Figs. 6 and 7 are still present also at larger L. This makes us confident with the fact that the sub-diffusive behavior survives in the thermodynamic limit. We have also modified the caption of Fig. 9 adding a comment about the apparent super-diffusive behavior.

4-Is there a reason to not include the curves for p=1 in, for example, Fig. 4. It could be useful to have that limit as well.

The reason why we did not put in the figures p=1 is to preserve the readability of the data. In fact, in order to include the values for p=1, we would need to change considerably the y-axis scale.

5-Figure 5. What is being plotted in the color scale and what is the scale. The caption seems to suggest that something binary is being plotted, but the white region would suggest otherwise.

We have added a bar with the color scale and extended the captions of both figs. 5 and 9 to provide a clearer explanation of these figures.

6-In the text explaining Figure 7, the authors discuss what happens at small p, but no data is provided in the case of small p. Would it be useful to include that data? Also the caption to the figure simply says "Comparison" but not comparison of what. Finally: the authors discuss how the two models are qualitatively the same after introducing a peff, yet in some cases a peak in one model gives a valley in the other. Not sure how seriously one should take this comparison.

Small p for us means in fact that p is small enough to be in the diffusive regime. In this sense, p=0.7 is considered as small.

Concerning the second part of the question, the idea of introducing peff is just to increase the qualitative resemblance between the two models. In particular we observe in figure 7 a very strong correlation between the positions of the resonances in the MDB and in the GOE protocols once p is “renormalized” to peff. Yet, We agree with the referee on the fact that at present we don’t know whether the two models are exactly mappable onto each other (and most likely they are not). In fact we mention in the text that the GOE bath is less effective in thermalizing the system with respect to the Limbladian dephasing.

7-Maybe add also γi to Figure 7?

We tried to add the \gamma_i to the figures but we did not find a way to do it preserving their readability.

8-Since the first two authors are co-first authors, it would maybe make sense to mark that somehow also in the author list, not only in the acknowledgment.

We have specified this just below the author’s list.

9-Appendix A: The authors fix J=1, likely this should be t=1.

Yes, we thank the referee for pointing out this typo, which we have fixed.

10-Notation in Eq. (19). Doesn't P depend on C?

Yes, the referee is right. We have specified that P depends on C.

11-Appendix B could be made a bit more self contained by a few more sentences of introduction. What are the boson variables, what is the starting point, etc. The could explain more clearly what loops they are referring to when they neglect all loops. And Gvi is not defined. This notation is used in their earlier work but not defined here.

We have amended App B to make it more self contained. We have added a new paragraph at the beginning to define the integration variables, and a new paragraph on page 24 to better define the cavity fields (including Gvi).

12-There are several references that are not complete. This seems to be mainly arXiv references that are missing the arXiv numbers.

We have fixed these issues.

Lastly, we have fixed all the minor typos and issues, including:

9-Is there an extra minus sign in Eq. (48)?

Yes, indeed the referee is right, the equation was incorrect. We have fixed the problem in the new version (p -> log p).

---

## Round 3 · List of Changes

In an unordered fashion: - Minor typos/ issues specified by the referee have been adjusted - Solved bibliographical incomplete entries - Implemented larger discussion in Appendix B - Fixed various notational issues (P depends on C, t=1 instead of J=1, etc.) - Added equal contribution of first co-authors in the preamble - Updated figures Fig 5 and Fig 9 with clearer explanations - Updated caption in Fig 7 - Added discussion on fitting procedure - Clarified discussion after Eq.14 regarding random regular graphs - Added sentence after Eq.(11) to clarify the energy dependence of the GOE case - Added a discussion at the end of Sec II concerning the comments of both referees

---

## Editorial Decision

published